# PROBLEM-PARAMETER-AGNOSTIC MAML

## ABSTRACT

Meta-learning aims to equip artificial intelligence systems with the ability to learn how to learn. Among various methods, Model-Agnostic Meta-Learning (MAML) is particularly effective for enabling rapid task adaptation. However, vanilla MAML suffers from a critical drawback: its performance is highly sensitive to carefully tuned hyperparameters, particularly the learning rate. Since these learning rates theoretically depend on problem-specific factors (e.g., task heterogeneity and loss smoothness), which are typically unknown, this reliance hinders training stability and limits adaptation performance. To address this challenge, we propose TFMAML, a tuning-free MAML algorithm that integrates adaptive stepsize and momentum techniques. TFMAML offers two key advantages: (i) it eliminates dependence on problem-specific parameters, allowing stepsizes to be pre-set without costly manual tuning or additional training process; ii) It guarantees convergence, unlike vanilla MAML, which lacks convergence guarantees to first-order stationary points. We provide rigorous theoretical analysis showing that TFMAML achieves the state-of-the-art convergence rate of $\mathcal{O}(\epsilon^{-4})$ to reach FOSP. Furthermore, we prove that its first-order variant, TFFOMAML, avoids Hessian computations while retaining the same $\mathcal{O}(\epsilon^{-4})$ convergence rate. Unlike standard First-Order MAML, which suffers from a constant error floor, TFFO-MAML eliminates this bias and converges reliably to stationary points. Extensive experiments validate our theory, demonstrating the clear superiority of TFMAML and TFFOMAML over existing benchmarks.

## 1 INTRODUCTION

### 1.1 BACKGROUND

Contemporary machine learning models are typically trained independently from scratch for specific tasks using manually designed, fixed learning algorithms (Hospedales et al., 2021). In contrast, meta-learning—often described as "learning to learn" (Thrun & Pratt, 1998)—aims to acquire transferable knowledge across tasks. By learning from a distribution of related tasks, a meta-learner extracts common structures and adaptation strategies, enabling faster and more data-efficient learning on novel tasks. Among various meta-learning approaches, the Model-Agnostic Meta-Learning (MAML) algorithm is one of the most widely adopted methods (Finn et al., 2017). Within the MAML framework, a Deep Neural Network (DNN) is trained across a diverse set of tasks to finds an initialization of parameters that can be rapidly adapted to new target tasks using only a small number of samples.

MAML has been widely adopted across diverse domains due to its simplicity and model-agnostic design. This popularity has led to numerous MAML-based variants (Qin et al., 2023; Kim & Hospedales, 2025; Zintgraf et al., 2019; Rajeswaran et al., 2019; Antoniou et al., 2018; Raghu et al., 2019). In particular, Finn et al. (2017) proposed a variant called First-Order Model-Agnostic Meta-Learning (FOMAML), which ignores second-order derivative terms, reducing computational complexity at the cost of losing some higher-order information. Furthermore, MAML has demonstrated success across various domains. Notable applications include few-shot natural language processing tasks (Yin, 2020; Hospedales et al., 2021) and few-shot image classification (Jeong & Kim, 2020; Ye & Chao, 2022).

Despite its broad applications, training the MAML algorithm faces several inherent challenges: (i) **Complex dual-loop optimization.** The bi-level structure of MAML requires backpropagation

through both inner and outer loops, involving higher-order derivatives. This often results in highly variable gradient scales (Antoniou et al., 2018), leading to unstable training. (ii) **Stochastic and biased gradients.** The few-shot adaptation setting and small-batch training introduce substantial stochasticity into gradient estimates. Moreover, the inner-loop fine-tuning step induces bias in the outer-loop gradient, complicating both optimization dynamics and theoretical convergence analysis (Fallah et al., 2020a). (iii) **Dependence on problem-specific step sizes.** The convergence of MAML theoretically depends on carefully chosen learning rates, which are determined by problem-specific quantities such as function smoothness and variance at both sample and task levels (Fallah et al., 2020a; Ji et al., 2022). These quantities are typically unknown in practice.

These challenges underscore the critical role of learning rate selection in stable MAML training. Existing approaches fall into two categories: (i) **Learning-based approaches.** One line of research learns learning rates through auxiliary neural networks (Antoniou et al., 2018; Li et al., 2017; Baik et al., 2020; Sun & Gao, 2023). For instance, Baik et al. (2020) proposed a hyperparameter generator network to control parameter update direction and magnitude. (ii) **Theory-driven approaches.** Another line of research determines learning rates in a training-free manner, grounded in convergence theory. Fallah et al. (2020a;b) established convergence guarantees for one-step MAML, showing that learning rates depend on task smoothness and variance properties. Ji et al. (2022) extended this analysis to multi-step inner-loop MAML, further revealing the dependence on problem-specific characteristics.

## 1.2 MAIN CONTRIBUTIONS

This paper addresses the critical limitations of MAML training by proposing tuning-free MAML (TFMAML), a novel approach that ensures stable convergence under stochastic and biased gradients while eliminating the need for problem-specific hyperparameter tuning. Our method combines training-free adaptive learning rate normalization with momentum-based stabilization. The adaptive learning rate mechanism automatically scales updates to account for gradient variance, while the momentum component mitigates instability from higher-order derivatives and accelerates convergence in the nonconvex landscape. We extend this method to MAML's first-order variant, FO-MAML, developing tuning-free FOMAML (TFFOMAML) with convergence guarantees. To our knowledge, TFMAML is the first algorithm to achieve parameter-free adaptation in MAML with provable convergence guarantees. Our main contributions are summarized as follows:

1. We introduce TFMAML, a tuning-free algorithm for nonconvex meta-learning that leverages adaptive learning rate normalization and momentum. TFMAML offers several key advantages:

   - **Independence from problem-specific parameters:** Unlike existing theory-driven approaches (Fallah et al., 2020a;b), TFMAML's learning rates do not require knowledge of smoothness constants, data variance, or task variance. All learning rates are determined explicitly by iteration counts, ensuring practical applicability.
   - **Stability under stochasticity and bias:** TFMAML maintains robustness against the stochasticity of few-shot tasks and the bias introduced by inner-loop fine-tuning, enabling stable meta-gradient updates.

2. We provide a rigorous theoretical analysis of TFMAML, demonstrating that it achieves state-of-the-art complexity for nonconvex problems. TFMAML achieves a convergence rate of $\mathcal{O}(1/T^{\frac{1}{4}})$ for the expected gradient norm under constant learning rates. Unlike existing approaches (Fallah et al., 2020a; Ji et al., 2022; Fallah et al., 2020b), **this convergence rate is not degraded by stochasticity-related constants**. Moreover, we prove that the first-order variant, TFFOMAML, maintains this $\mathcal{O}(1/T^{\frac{1}{4}})$ convergence rate, **resolving the non-vanishing bias issue** inherent in previous first-order approximations (Fallah et al., 2020a). Our analysis requires only standard assumptions of MAML (Fallah et al., 2020a; Ji et al., 2022; Fallah et al., 2020b) and makes no assumptions about convexity (Chayti & Jaggi, 2024; Yang et al., 2025).

3. We conduct extensive evaluations on standard few-shot learning benchmarks, with strictly separated training and test sets, comparing TFMAML, TFFOMAML to vanilla MAML under various optimizers over various datasets.The results demonstrate improved stability and robustness to learning-rate choices, validating both our theoretical guarantees and practical effectiveness.

## 2 PROBLEM SETUP

### 2.1 MODEL-AGNOSTIC META LEARNING

We consider the standard multi-task learning framework with task index set $\mathcal{I} = \{1, \ldots, I\}$, where each task $i \in \mathcal{I}$ is associated with data distribution $\mathcal{P}_i$. The population risk minimization problem is formulated as:

$$\min_{\boldsymbol{\theta} \in \mathbb{R}^d} f(\boldsymbol{\theta}) := \mathbb{E}_i\big[f_i(\boldsymbol{\theta})\big], \text{ s.t. } f_i(\boldsymbol{\theta}) := \mathbb{E}_{\mathcal{D} \sim \mathcal{P}_i}\big[f_i(\boldsymbol{\theta}; \mathcal{D})\big]$$

where $f_i(\cdot)$ denotes the loss function for task $i$.

Unlike conventional multi-task learning, MAML (Finn et al., 2017) seeks to learn an initialization that enables rapid adaptation to new tasks through few gradient descent steps. For clarity, we consider a single inner-loop update[1], yielding the MAML objective $F(\boldsymbol{\theta}) := \mathbb{E}_i[f_i(\boldsymbol{\theta} - \alpha \nabla f_i(\boldsymbol{\theta}))]$, with corresponding gradient given by:

$$\nabla F(\boldsymbol{\theta}) = \frac{1}{I} \sum_{i \in \mathcal{I}} \left(\mathbf{I} - \alpha \nabla^2 f_i(\boldsymbol{\theta})\right) \nabla f_i(\boldsymbol{\theta} - \alpha \nabla f_i(\boldsymbol{\theta})). \tag{1}$$

In practice, since the total number of tasks $I$ can be large and each task provides only limited data, only a small subset of tasks and samples are accessible in each iteration. Let $\mathcal{I}_t \subset \mathcal{I}$ denote the task index set sampled at iteration $t$, with $|\mathcal{I}_t| = J \ll I$. For each task $j \in \mathcal{I}_t$, the dataset $\mathcal{D}_j$ is split into a *support set* $\mathcal{D}_j^{spt}$ for task-specific finetuning, and a *query set* $\mathcal{D}_j^{qry}$ for validating the updated model's performance, where typically $|\mathcal{D}_j^{spt}| \ll |\mathcal{D}_j^{qry}|$. We assume uniform sample numbers across tasks, denoting $D = |\mathcal{D}_j|$, $D^{spt} = |\mathcal{D}_j^{spt}|$, and $D^{qry} = |\mathcal{D}_j^{qry}|$. The empirical support and query losses are defined as $f_j(\boldsymbol{\theta}; \mathcal{D}_j^{spt}) := \frac{1}{D^{spt}} \sum_{z \in \mathcal{D}_j^{spt}} f_j(\boldsymbol{\theta}; z)$ and $f_j(\boldsymbol{\theta}; \mathcal{D}_j^{qry}) := \frac{1}{D^{qry}} \sum_{z \in \mathcal{D}_j^{qry}} f_j(\boldsymbol{\theta}; z)$, respectively. At iteration $t$, **MAML** performs two levels of updates:

**Inner loop:** For each task $j \in \mathcal{I}_t$, the task-specific parameter $\boldsymbol{\phi}_j$ is obtained by one gradient step on the support set with learning rate $\alpha$: $\boldsymbol{\phi}_j = \boldsymbol{\theta} - \alpha \nabla f_j(\boldsymbol{\theta}; \mathcal{D}_j^{spt})$.

**Outer loop:** The shared parameter $\boldsymbol{\theta}$ is then updated using the average loss over the query sets from the sampled tasks. With learning rate $\beta$, the update is: $\boldsymbol{\theta} \leftarrow \boldsymbol{\theta} - \beta \cdot \frac{1}{J} \sum_{j \in \mathcal{I}_t} \nabla_{\boldsymbol{\theta}} f_j(\boldsymbol{\phi}_j; \mathcal{D}_j^{qry})$. The corresponding meta-gradient is $\nabla \hat{F}(\boldsymbol{\theta}) = \nabla_{\boldsymbol{\theta}} \left(\frac{1}{J} \sum_{j \in \mathcal{I}_t} f_j\big(\boldsymbol{\theta} - \alpha \nabla f_j(\boldsymbol{\theta}; \mathcal{D}_j^{spt}); \mathcal{D}_j^{qry}\big)\right)$, which, by the chain rule, expands to

$$\nabla \hat{F}(\boldsymbol{\theta}) = \frac{1}{J} \sum_{j \in \mathcal{I}_t} \left(\mathbf{I} - \alpha \nabla^2 f_j(\boldsymbol{\theta}; \mathcal{D}_j^{spt})\right) \nabla f_j\big(\boldsymbol{\theta} - \alpha \nabla f_j(\boldsymbol{\theta}; \mathcal{D}_j^{spt}); \mathcal{D}_j^{qry}\big). \tag{2}$$

A comparison between this stochastic meta-gradient eq. (2) and the ideal full-batch gradient eq. (1) reveals two primary sources of stochasticity:

1. **Support and query set variability:** The inner-loop update uses the randomly drawn support set $\mathcal{D}_j^{spt}$, affecting both the gradient and Hessian calculations. The outer-loop update similarly uses the randomly sampled query set $\mathcal{D}_j^{qry}$.

2. **Task sampling variability:** At each iteration, the task subset $\mathcal{I}_t$ of size $J$ is randomly sampled from the full task set $\mathcal{I}$.

These stochasticity will cause challenges to the convergence of the vanilla MAML algorithm, as will be detailed in section 4.1.

A widely used variant, **FOMAML** (Finn et al., 2017; Nichol et al., 2018), approximates the meta-gradient by dropping the second-order term (equivalently, treating the inner update as a stop-gradient). The resulting estimator is

$$\nabla \hat{F}_1(\boldsymbol{\theta}) = \frac{1}{J} \sum_{j \in \mathcal{I}_t} \nabla f_j\big(\boldsymbol{\theta} - \alpha \nabla f_j(\boldsymbol{\theta}; \mathcal{D}_j^{spt}); \mathcal{D}_j^{qry}\big). \tag{3}$$

This first-order approximation avoids Hessian-vector products and reduces computation, at the cost of additional bias relative to eq. (2).

---

[1]Single-step updates are standard in practice and sufficient to illustrate the key concepts.

## 2.2 BASIC ASSUMPTIONS

We adopt the following assumptions, which are standard in the meta-learning literature (Fallah et al., 2020a;b; Ji et al., 2022; Yang & Kwok, 2024).

**Assumption 1** (Bounded Loss). *$F$ is bounded below, $\min_{\boldsymbol{\theta}} F(\boldsymbol{\theta}) > -\infty$ and the optimality gap $\Delta := F(\boldsymbol{\theta}) - \min_{\boldsymbol{\theta} \in \mathbb{R}^d} F(\boldsymbol{\theta})$ is bounded for any $\boldsymbol{\theta} \in \mathbb{R}^d$.*

**Assumption 2** (Smoothness). *For any $i \in \mathcal{I}$, there exists constants $l$, $G$, and $\rho$ such that*

$$\|\nabla f_i(w) - \nabla f_i(u)\| \le l\|w - u\|, \ \|\nabla f_i(w)\| \le G. \tag{4}$$

$$\|\nabla^2 f_i(w) - \nabla^2 f_i(u)\| \le \rho\|w - u\| \text{ for any } w, u. \tag{5}$$

**Assumption 3** (Bounded Variance with Task Sampling). *There exists constants $\sigma_G, \sigma_H > 0$, such that*

$$\mathbb{E}_i\left[\|\nabla f_i(w) - \nabla f(w)\|^2\right] \le \sigma_G^2, \text{ for any } w.$$

$$\mathbb{E}_i\left[\|\nabla^2 f_i(w) - \nabla^2 f(w)\|^2\right] \le \sigma_H^2, \text{ for any } w.$$

*Note that under Assumption 2, the conditions in Assumption 3 are automatically satisfied for $\sigma_G = 2G$ and $\sigma_H = 2l$. However, we state this assumption separately to highlight the role of similarity of functions corresponding to different tasks in convergence analysis.*

**Assumption 4** (Bounded Variance with Data Sampling). *For any $i \in \mathcal{I}$, there exists constants $\tilde{\sigma}_G, \tilde{\sigma}_H > 0$, such that*

$$\mathbb{E}_{\mathcal{D} \sim \mathcal{P}_i}\left[\|\nabla f_i(w; \mathcal{D}) - \nabla f_i(w)\|^2\right] \le \tilde{\sigma}_G^2, \text{ for any } w.$$

$$\mathbb{E}_{\mathcal{D} \sim \mathcal{P}_i}\left[\|\nabla^2 f_i(w; \mathcal{D}) - \nabla^2 f_i(w)\|^2\right] \le \tilde{\sigma}_H^2, \text{ for any } w.$$

## 3 PROPOSED ALGORITHMS

### 3.1 MAIN CHALLENGES

Before presenting our tuning-free MAML algorithm, we first identify key challenges that prevent standard MAML and FOMAML from achieving convergence. In practice, computing the ideal full-batch gradient (eq. (1)) is intractable due to the large number of tasks and second-order derivative computations. For efficiency, we typically use the simplified gradient estimator (eq. (2)), but this approximation introduces both data and task sampling stochasticity. Specifically,

- **The descent direction in MAML is a biased estimator of $\nabla F(\boldsymbol{\theta})$.** For a given initialization $\boldsymbol{\theta}$, the empirical gradient $\nabla \hat{F}(\boldsymbol{\theta})$ (eq. (2)) is biased relative to the ideal gradient $\nabla F(\boldsymbol{\theta})$ (eq. (1)), since the support-set gradient is nested within the query-set gradient. This inner-loop adaptation induces a systematic bias. Moreover, under suitable conditions, **momentum can serve as an effective technique to stabilize biased or noisy estimates**, both in MAML/FOMAML (Mai & Johansson, 2020).

- **Problem-dependent smoothness and stepsize selection.** In MAML, the global objective $F$ inherits a nested inner–outer structure across tasks, making its smoothness parameter depend on task-specific gradients, Hessians, and gradient norms. Since optimization methods such as SGD scale the stepsize inversely with smoothness, this variability complicates stepsize selection. In practice, these parameters are inaccessible due to the complexity of DNNs and task heterogeneity. **Fortunately, adaptive stepsize schemes offer a practical remedy (Yang et al., 2024).**

### 3.2 ALGORITHM DEVELOPMENT OF TFMAML

Motivated by the above challenges, we design TFMAML to couple momentum with gradient normalization so that: (i) Momentum-induced exponential averaging attenuates bias, and (ii) The adaptive step size adapts to the global gradient magnitude. Concretely, we focus on directly leverage gradients to adjust the stepsize by incorporate MAML into the normalized stochastic gradient descent with momentum (NSGD-M) framework (Yang et al., 2023; Cutkosky, 2023), defined as:

$$\begin{cases} \boldsymbol{g}_{t+1} = (1 - \gamma)\boldsymbol{g}_t + \gamma \nabla \hat{F}(\boldsymbol{\theta}_{t+1}), \\ \boldsymbol{\theta}_{t+1} = \boldsymbol{\theta}_t - \frac{\beta}{\|\boldsymbol{g}_t\|} \cdot \boldsymbol{g}_t, \end{cases} \tag{6}$$

---

**Algorithm 1:** MAML Algorithm with Normalized Momentum Update

---

**Input:** Training tasks $\mathcal{D}_i, i \in \{1, \cdots, I\}$ ; Inner loop step size $\alpha$; Outer loop step size $\beta$;
Momentum parameter $\gamma$; Number of iterations $T$
**Output:** Meta-trained network initialization $\boldsymbol{\theta}_T$

---

1  *Meta-training stage*
2  Initialize $\boldsymbol{\theta}$ randomly.
3  **for** $t \in \{0, \cdots, T-1\}$ **do**
4      Uniformly sample $J$ meta-training tasks with index set $\mathcal{I}_t$.
5      **forall** $j \in \mathcal{I}_t$ **do**
6          Initialize the network parameter $\boldsymbol{\theta}_t$.
7          Uniformly sample $D^{spt}$ data samples from $\mathcal{D}_j$ as the support set.
8          Draw another $D^{qry}$ samples from $\mathcal{D}_j$ as the query set with $D^{qry} \gg D^{spt}$.
9          Evaluate inner loop loss $f_j(\boldsymbol{\theta}_t; \mathcal{D}_j^{spt})$ on support set $\mathcal{D}_j^{spt}$. Update the task-specific
          parameters by gradient descent: $\boldsymbol{\phi}_j = \boldsymbol{\theta}_t - \alpha \nabla f_j(\boldsymbol{\theta}_t; \mathcal{D}_j^{spt})$.
10         Evaluate loss $f_j(\boldsymbol{\phi}_j; \mathcal{D}_j^{qry})$ on query set $\mathcal{D}_j^{spt}$.
11     Compute outer loop global gradient: $\nabla \hat{F}(\boldsymbol{\theta}_{t+1}) = \frac{1}{J} \sum_{j \in \mathcal{I}_t} \left( \mathbf{I} - \alpha \nabla^2 f_j(\boldsymbol{\theta}_t; \mathcal{D}_j^{spt}) \right) \cdot \nabla f_j \left( \boldsymbol{\phi}_j; \mathcal{D}_j^{qry} \right).$
12     Perform outer loop momentum update: $\boldsymbol{g}_{t+1} = (1-\gamma)\boldsymbol{g}_t + \gamma \nabla \hat{F}(\boldsymbol{\theta}_{t+1}).$
13     Perform normalized outer loop update: $\boldsymbol{\theta}_{t+1} \leftarrow \boldsymbol{\theta}_t - \frac{\beta}{\|\boldsymbol{g}_{t+1}\|} \boldsymbol{g}_{t+1}.$

14 *Meta-finetuning stage*
15 Load the initialized network parameter $\boldsymbol{\theta}_T$.
16 Sample $D$ data points from new tasks $\mathcal{D}_{test}$ as the support set $\mathcal{D}_{test}^{spt}$ for finetuning. The
   remaining data of $\mathcal{D}_{test}$ consists query set $\mathcal{D}_{test}^{qry}$.
17 Evaluate loss $f_{test}(\boldsymbol{\theta}_T; \mathcal{D}_{test}^{spt})$ on support set $\mathcal{D}_{test}^{spt}$. Finetune the parameters via gradient
   descent as $\boldsymbol{\theta}_{test} \leftarrow \boldsymbol{\theta}_T - \alpha \nabla f_{test}(\boldsymbol{\theta}_T; \mathcal{D}_{test}^{spt})$.

---

with momentum initialized as $\boldsymbol{g}_0 = \nabla \hat{F}(\boldsymbol{\theta}_0)$. This global update rule reflects several important intuitions:

- **Momentum aggregates historical gradients.** The momentum term effectively performs an exponential moving average over past outer loop gradients. Concretely, the update can be expressed as

$$\boldsymbol{g}_{t+1} = (1-\gamma)\boldsymbol{g}_t + \gamma \Big( \frac{1}{J} \sum_{j \in \mathcal{I}_t} \left( \mathbf{I} - \alpha \nabla^2 f_j(\boldsymbol{\theta}_t; \mathcal{D}_j^{spt}) \right) \cdot \nabla f_j \left( \boldsymbol{\phi}_j; \mathcal{D}_j^{qry} \right) \Big),$$

Here, the small value of $\gamma$ ensures stability by preventing large deviations caused by high-variance gradient estimates in a single iteration. This formulation effectively accumulates descent directions across iterations and tasks, smoothing the trajectory of optimization. By incorporating gradients from multiple tasks over time, the momentum buffer helps mitigate the effects of task heterogeneity and data sampling noise. As a result, the optimization process becomes more robust and less sensitive to fluctuations in individual task updates.

- **Normalized gradient update.** To ensure stability across iterations, we adopt an adaptive stepsize by normalizing the descent direction $\boldsymbol{g}_t$, effectively using a fixed-length update vector $\beta \cdot \boldsymbol{g}_t / \|\boldsymbol{g}_t\|$. Since each iteration involves task-specific updates over randomly sampled tasks and data batches, the gradient magnitude can vary significantly due to task heterogeneity. Normalization ensures consistent update magnitudes across iterations, preventing any batch of tasks or samples from disproportionately influencing the global step (Yan et al., 2025). This approach also mitigates scenarios where gradient magnitudes become excessively large—an effect akin to stepsize adjustment in works such as (Fallah et al., 2020a; Yang & Kwok, 2024). Unlike those methods, which adapt stepsizes based on gradient norms and problem-specific parameters, we employ a simpler normalization strategy that is task-agnostic and implementation-friendly. Moreover, normalized updates simplify the theoretical analysis by ensuring that the distance

between consecutive iterates is constant in $\ell_2$-norm

$$\|\boldsymbol{\theta}_{t+1} - \boldsymbol{\theta}_t\| = \left\| \beta \frac{\boldsymbol{g}_t}{\|\boldsymbol{g}_t\|} \right\| = \beta. \tag{7}$$

Furthermore, due to the elimination of Hessian matrix, the gradient update direction of FOMAML is highly biased compared to the ideal gradient in eq. (1). Similarly, we conduct gradient normalization and momentum aggregation as in algorithm 2 in section B.4. Specifically, the aggregated momentum can be written as

$$\boldsymbol{g}_{t+1} = (1 - \gamma)\boldsymbol{g}_t + \gamma \Big( \frac{1}{J} \sum_{j \in \mathcal{I}_t} \nabla f_j \left( \boldsymbol{\phi}_j; \mathcal{D}_j^{qry} \right) \Big).$$

# 4 THEORETICAL RESULTS AND COMPARISONS WITH PRIOR WORKS

## 4.1 INHERENT STOCHASTICITY IN MAML

Establishing problem-parameter-agnostic convergence guarantees for MAML is difficult due to its coupled inner–outer update structure. As analyzed in Fallah et al. (2020a), two fundamental obstacles arise:

1. the stochastic inner update shifts the evaluation point of the outer objective, producing an inherently **biased** meta-gradient direction; and

2. the global meta-gradient depends jointly on task-level first- and second-order smoothness as well as the chosen inner stepsize, yielding a **complex, unknown effective smoothness**.

Because of these effects, classical convergence analyses, which is built on unbiased stochastic gradients and known smoothness constants, do not directly apply. Importantly, Fallah et al. (2020a) characterizes these difficulties but does not provide a mechanism to overcome them without invoking problem-specific parameters. In contrast, our main contribution is to develop a simple, **problem-parameter-free** meta-optimizer that **provably** handles both sources of difficulty simultaneously. For clarity, we formalize these two challenges through lemma 1 and lemma 2.

**Lemma 1** (Lipschitz smoothness parameter of the global gradient). *Consider the global gradient* $\nabla F$ *as eq.* (1). *For any* $w, u \in \mathbb{R}^d$, *we have*

$$\|\nabla F(w) - \nabla F(u)\| \le L \|w - u\|, \tag{8}$$

*where the global objective function's smoothness parameter is*

$$L := \min_{w,u} \{L(w), L(u)\},$$

$$L(w) := l(1 + \alpha l)^2 + \rho\alpha(1 + \alpha l)\mathbb{E}_i \left[ \|\nabla f_i(w)\| \right].$$

**Remark 1.** *The dual-loop structure of MAML induces a complex smoothness parameter:* *When* $\alpha = 0$, *MAML reduces to standard multi-task learning with smoothness constant* $L = l$ *(Assumption 2). In contrast, full MAML couples inner- and outer-loop updates, yielding a global objective* $F(\boldsymbol{\theta})$ *whose smoothness depends jointly on the gradient Lipschitz constant* $l$ *and the Hessian Lipschitz constant* $\rho$, *making the global-gradient* $\nabla F(\boldsymbol{\theta})$ *considerably more intricate.*

**Remark 2.** *The smoothness parameter is difficult to compute:* *Lemma 1 shows that the global objective function* $F$ *is smooth, but with a parameter that depends on the minimum expected gradient norm. Specifically, for some classical operation of evaluating smoothness between two points* $w$ *and* $u$ *(Fallah et al., 2020a;b; Ji et al., 2022), the bound involves* $\min \{\mathbb{E}_i \|\nabla f_i(w)\|, \mathbb{E}_i |\nabla f_i(u)|\}$, *which is costly to compute in practice since it requires accessing gradients across all tasks.*

Another key difficulty in analyzing MAML lies in the fact that the stochastic global-gradient $\nabla \hat{F}(\boldsymbol{\theta})$ is a *biased* estimator of the true global-gradient $\nabla F(\boldsymbol{\theta})$. The bias arises because the outer-loop gradient is evaluated at an adapted parameter obtained from a stochastic inner-loop update. Concretely, the update $\nabla f_j \left( \boldsymbol{\theta} - \alpha \nabla f_j(\boldsymbol{\theta}; \mathcal{D}_j^{spt}); \mathcal{D}_j^{qry} \right)$ depends on both a randomly sampled support set $\mathcal{D}_j^{spt}$ and query set $\mathcal{D}_j^{qry}$. This nested stochasticity couples the inner and outer updates, invalidating the unbiasedness assumption commonly used in SGD analysis. We formalize this as follows.

**Lemma 2** (**Biased Global-Gradient**). *The stochastic outer-loop gradient in eq. (2) is a biased estimator of $\nabla F(\boldsymbol{\theta})$ (eq. (1)). Specifically, for any task $j$,*

$$\mathbb{E}_{\mathcal{D}_j^{spt}, \mathcal{D}_j^{qry}} \left[ \nabla f_j \big( \boldsymbol{\theta} - \alpha \nabla f_j(\boldsymbol{\theta}; \mathcal{D}_j^{spt}); \mathcal{D}_j^{qry} \big) \right] = \nabla f_j \big( \boldsymbol{\theta} - \alpha \nabla f_j(\boldsymbol{\theta}) \big) + \boldsymbol{e}_j, \quad \|\boldsymbol{e}_j\| \leq \frac{\alpha l \tilde{\sigma}_G}{\sqrt{D^{spt}}},$$

*Moreover, averaging across tasks,*

$$\mathbb{E}_{j, \mathcal{D}_j^{spt}, \mathcal{D}_j^{qry}} \left[ \nabla f_j \big( \boldsymbol{\theta} - \alpha \nabla f_j(\boldsymbol{\theta}; \mathcal{D}_j^{spt}); \mathcal{D}_j^{qry} \big) \right] = \mathbb{E}_j [\nabla f_j \big( \boldsymbol{\theta} - \alpha \nabla f_j(\boldsymbol{\theta}) \big)] + \boldsymbol{e}, \quad \|\boldsymbol{e}\| \leq \frac{\alpha l \tilde{\sigma}_G}{\sqrt{D^{spt} J}}.$$

*Hence, the descent direction used in the MAML update $\nabla \hat{F}(\boldsymbol{\theta})$ is systematically biased, posing an additional challenge for establishing its convergence guarantees.*

Building on Lemmas lemma 1 and lemma 2—which highlight the complexity of the global gradient's smoothness and the bias in the MAML gradient —we observe the challenge of establishing convergence guarantees for MAML (Fallah et al., 2020a;b). In particular, the stepsize is tightly coupled with problem-dependent parameters, including the smoothness constants $l$, $\rho$, and the expected gradient norm $\mathbb{E}_i [\|\nabla f_i(\boldsymbol{\theta})\|]$. Estimating these quantities is computationally expensive, rendering stepsize tuning highly nontrivial. More critically, the gradient bias may prevent MAML from converging to an $\epsilon$-stationary point (Fallah et al., 2020a), even with carefully chosen stepsizes. To overcome these issues, as state in section 3.2, we couple momentum with gradient normalization in the global-update. This decays sampling-induced global-gradient bias and adaptively rescales updates, yielding tuning-free inner and outer step sizes. Consequently, the method converges to a first-order stationary point without pre-estimating problem-dependent parameters, as shown by the theorems in the next subsection.

## 4.2 THEORETICAL RESULTS

In this subsection, we present the main theoretical guarantees for the proposed tuning-free MAML and tuning-free FOMAML algorithms. By combining momentum-augmented global updates with adaptive stepsize schedules, we establish in the theorems below that our methods converge to a first-order stationary point while remaining entirely tuning-free—i.e., they require no adjustment based on unknown problem-dependent constants such as smoothness or curvature.

**Theorem 1** (**Tuning-Free MAML with Convergence Guarantee**). *Suppose that Assumptions $2 \sim 4$ hold. Let the inner and outer-loop stepsize be $\alpha = \frac{\sqrt[4]{D^{spt}}}{T^{\frac{1}{4}}}$, $\beta = \frac{D^{qry} \times J}{T^{\frac{3}{4}}}$ and momentum decaying weight as $\gamma = \frac{D^{qry} \times J}{T^{\frac{1}{2}}}$. The model model parameter $\{\boldsymbol{\theta}_t\}_{t=0}^{T}$ are iteratively generated by Algorithm 1. Then, it holds for all $T \leq 1$ that*

$$\frac{1}{T} \sum_{t=0}^{T-1} \mathbb{E}\left[\|\nabla F(\boldsymbol{\theta}_t)\|\right] \leq \mathcal{O}\left( \frac{\mathcal{C}_1}{T^{\frac{1}{4}}} + \frac{\mathcal{C}_2}{T^{\frac{1}{2}}} \right),$$

*where $\mathcal{C}_1 = \max\{\frac{\Delta}{D^{qry}J}, l + \sqrt{D^{qry}}\tilde{\sigma}_G \sigma_G, \frac{l\tilde{\sigma}_G}{\sqrt[4]{D^{spt}}}\}$ and $\mathcal{C}_2 = \max\{(l^2 + 1)\tilde{\sigma}_G + \sqrt{D^{qry}}\sqrt[4]{D^{spt}} \left( \sigma_H G + l\sigma_G + \sigma_G \sigma_H \right), \frac{\sqrt{D^{qry}}G\tilde{\sigma}_H + \sqrt{D^{qry}}l\tilde{\sigma}_G + \tilde{\sigma}_H \tilde{\sigma}_G}{\sqrt[4]{D^{spt}}}, \frac{\tilde{\sigma}_G + \sqrt{D^{qry}}\sigma_G}{(D^{qry}J)^{3/2}}\}$.*

**Theorem 2** (**Tuning-Free FOMAML with Convergence Guarantee**). *Suppose that Assumptions $2 \sim 4$ hold. Let the inner and outer-loop stepsize be $\alpha = \frac{\sqrt[4]{D^{spt}}}{T^{\frac{1}{4}}}$, $\beta = \frac{D^{qry} \times J}{T^{\frac{3}{4}}}$ and momentum decaying weight as $\gamma = \frac{D^{qry} \times J}{T^{\frac{1}{2}}}$. The model model parameter $\{\boldsymbol{\theta}_t\}_{t=0}^{T}$ are iteratively generated by Algorithm 2. Then, it holds for all $T \leq 1$ that*

$$\frac{1}{T} \sum_{t=0}^{T-1} \mathbb{E}\left[\|\nabla F(\boldsymbol{\theta}_t)\|\right] \leq \mathcal{O}\left( \frac{\mathcal{C}_3}{T^{\frac{1}{4}}} + \frac{\mathcal{C}_4}{T^{\frac{1}{2}}} \right),$$

*where $\mathcal{C}_3 = \max\{\frac{\Delta}{D^{qry}J}, \tilde{\sigma}_G + l + \sqrt[4]{D^{spt}}lG, \frac{l\tilde{\sigma}_G}{\sqrt[4]{D^{spt}}}\}$ and $\mathcal{C}_4 = \max\{\sqrt{D^{qry}}\sqrt[4]{D^{spt}}lG, \frac{\sqrt{D^{qry}}\tilde{\sigma}_G}{\sqrt[4]{D^{spt}}}, \frac{\tilde{\sigma}_G}{(D^{qry}J)^{3/2}}\}$.*

**Remark 3** (**Problem-Parameter-Agnostic schedules for TFMAML**). *Existing global-learning methods often set learning rates based on unknown problem constants (e.g., smoothness or curvature), which vary across tasks and are hard to estimate. By contrast, the schedules in the theorems above are tuning-free: the step sizes $\alpha, \beta$ and momentum parameters $\gamma$ depend only on observable training knobs—the per-task support-set size $D^{spt}$, the per-task query-set size $D^{qry}$, the number of tasks sampled per iteration $J$, and the total horizon $T$. This yields a plug-and-play procedure across datasets and tasks, requiring no pre-calibration of hidden constants.*

### 4.3 COMPARISONS WITH PRIOR WORK

**Remark 4** (**On stepsize setting and convergence guarantees in vanilla MAML**). *In prior analyses of MAML, such as (Fallah et al., 2020a;b; Ji et al., 2022), the outer-loop stepsize is typically chosen in proportion to the inverse of a smoothness parameter. Since this parameter depends on gradient norms across all tasks, which are generally unknown and computationally intractable, existing works resort to stochastic estimates. For example, (Fallah et al., 2020a) approximates the smoothness constant by sampling a mini-batch of tasks and using the empirical average of their gradient norms: $\hat{L}(\boldsymbol{\theta}) := l(1 + \alpha l)^2 + \frac{\rho \alpha (1 + \alpha l)}{J} \sum_{j \in \mathcal{I}_t} \left\| \nabla f_j(\boldsymbol{\theta}; \mathcal{D}_j^{spt}) \right\|$, leading to the stochastic stepsize rule $\hat{\beta}(\boldsymbol{\theta}) = \frac{c}{\hat{L}(\boldsymbol{\theta})}$, where $\hat{L}(\boldsymbol{\theta})$ depends on both the sampled task set and the corresponding support datasets. By further setting $\alpha \in (0, \frac{1}{6l}]$, vanilla MAML finds a solution $\boldsymbol{\theta}$ such that $\mathbb{E}\left[ \|\nabla F(\boldsymbol{\theta})\| \right] \leq \mathcal{C} + \mathcal{O}(\frac{l}{T^{1/2}})$, where $\mathcal{C} = \sqrt{\frac{\sigma_G^2}{J} + \frac{\tilde{\sigma}_G^2}{JD^{qry}} + \frac{\tilde{\sigma}_G^2}{D^{spt}}}$ in a non-diminishing constant.*

*This approach highlights two key limitations of existing methods:*

1. *The **stepsize tuning** inevitably depends on problem-specific smoothness parameters;*

2. *The **convergence guarantees** include a non-vanishing constant error term, preventing true $\epsilon$-FOSP convergence control.*

**Our contribution:** We overcome both limitations by introducing a *tuning-free* variant of MAML based on *outer-loop gradient normalization* and *momentum aggregation*.

- **Problem-parameter–free step sizes.** Our algorithm provides a genuinely problem-parameter–agnostic step-size schedule, as formalized in lemma 1 and highlighted in Remark 3. This addresses a well-recognized difficulty in MAML optimization Fallah et al. (2020a); Ji et al. (2022). In contrast to prior SGD-based analyses Fallah et al. (2020a), whose step sizes depend on loss smoothness constants and gradient norms across tasks—quantities that are unknown and often intractable to estimate, our gradient normalization mechanism automatically adjusts the effective step size throughout training. The normalization dynamics are rigorously derived in Appendix B.2, lemma 3.

- **Sharper convergence with vanishing error.** We establish convergence to first-order stationary points without the residual error term that appears in existing MAML analyses (remark 4), as stated in Theorem 1. Our guarantee is both explicitly controllable and asymptotically vanishing in $T$. A key driver of this improvement is that momentum recursion progressively suppresses the meta-gradient bias (lemma 2), a mechanism quantified in Appendix B.2, lemma 4.

In summary, our method eliminates the need for problem-dependent hyperparameter tuning and delivers a provably tighter and more reliable convergence guarantee than existing MAML algorithms.

## 5 NUMERICAL RESULTS

To evaluate TFMAML against the non-adaptive baseline MAML (Finn et al., 2017), we conduct experiments on the Omniglot benchmark (Lake et al., 2011). Extended numerical results on mini-Imagenet dataset Ravi & Larochelle (2017) can be found in Appendix C.2. Omniglot consists of 1,623 handwritten characters from 50 alphabets, each represented by 20 instances produced by different individuals, and is widely regarded as a standard benchmark for few-shot image recognition

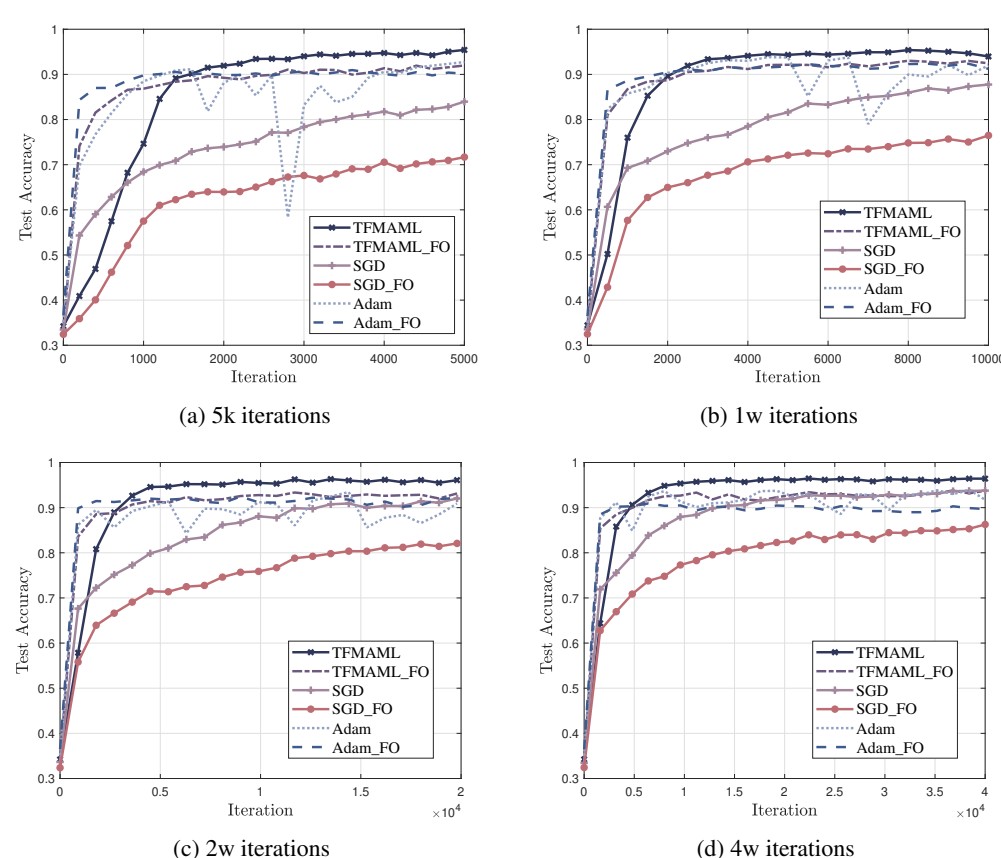

Figure 1: MAML and FOMAML's performance with different iteration setting. Notice that the gradients are calculated beforehand based on the pre-set number of iterations.

(Vinyals et al., 2016; Santoro et al., 2016). Following the conventional split, 1,200 characters are used for training and the remaining 423 for testing. We adopt the 5-way 1-shot classification setting, with 15 query examples per class. Each meta-update samples 20 tasks uniformly at random. The embedding network follows the canonical four-layer convolutional architecture of Vinyals et al. (2016), with all images resized to $28 \times 28$.

The inner- and outer-loop stepsizes for the benchmark methods follow their default configurations. Specifically, we use the open-source MAML implementation[2], where $\alpha = 0.4$ and $\beta = 10^{-3}$. While existing theoretical analyses of MAML and FOMAML (Fallah et al., 2020a;b; Ji et al., 2022) are derived under generic Stochastic Gradient Descent (SGD) updates, the original reference implementation (Finn et al., 2017) employs Adam in the outer loop. For completeness and fair comparison, we therefore include Adam as a benchmark in our experiments. In both the training and evaluation phases, we use a single gradient step for the inner-loop update. This choice not only keeps the computational cost minimal but also ensures methodological consistency with the assumptions underlying the theoretical results.

For our proposed methods, the stepsizes are determined by Theorem 1, where the iteration budget explicitly sets $\alpha$, $\beta$, and the momentum decay $\gamma$, resulting in distinct convergence trajectories. As shown in the figures, TFMAML exhibits slower initial convergence compared to SGD and Adam, but achieves superior asymptotic performance due to its theoretically grounded stepsize schedule and its no-constant-gap convergence guarantee.

---

[2]https://github.com/dragen1860/MAML-Pytorch

TFMAML, which incorporates second-order corrections, produces more accurate but higher-variance gradient estimates, often delaying stabilization. In contrast, TFFOMAML uses a first-order approximation, yielding smoother updates and faster per-iteration progress, albeit occasionally converging to slightly suboptimal final solutions. The instability of Adam arises from its adaptive scaling interacting with non-stationary, high-variance meta-gradients, whereas SGD and NSGD-M maintain consistent step sizes, with momentum effectively damping noise rather than amplifying curvature effects.

## 6 CONCLUSIONS

This paper has proposed an efficient approach to eliminating problem-specific parameter dependencies in MAML, enabling parameter-agnostic generalization across diverse settings. Our algorithms have also removed the need for data heterogeneity bounds and accommodated partial tasks participation, further broadening its applicability to real-world scenarios. We have provided a rigorous theoretical analysis demonstrating state-of-the-art convergence rate, based on the basic assumptions. Additionally, the proposed scheme ensures a convergence-guaranteed FOMAML, achieving the state-of-the-art $\mathcal{O}\left(\epsilon^{-4}\right)$. Furthermore, we have provided extensive numerical evidence to verify the efficacy of our approaches.

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

## A    RELATED WORKS

**Adaptive Stepsizes with Momentum.** Several works have combined adaptivity and momentum to improve convergence (Chen et al., 2020; Karimireddy et al., 2020; Wu et al., 2023), but these approaches still rely on tuned stepsizes and often impose restrictive assumptions. Even recent locally adaptive methods (Sohom Mukherjee, 2024) remain hyperparameter-dependent, requiring thresholds tied to problem smoothness. Hence, fully tuning-free adaptivity remains an open challenge.

**Problem-Parameter-Free Algorithms.** Some notable concurrent works Li et al. (2024); Yan et al. (2025) also explores problem-parameter-free algorithms in the context of decentralized non-convex optimization. While both studies target problem-parameter-free optimization, our work addresses the unique challenges inherent to MAML setting. The fundamental distinction lies in the inherent biased gradient and complex global smoothness parameter of MAML. The seemingly direct use of adaptive stepsize and momentum are jointly effective in face of these existing challenges.

**MAML-specific stepsize setups and convergence analysis** Sun & Gao (2023) incorporated momentum into the optimization process and meta-learn the learning rate, and using a LSTM-based DNN to learn the stepsizes and momentum decaying rate. Finn et al. (2019) analyzed online MAML for a strongly convex objective function. Fallah et al. (2020a;b) developed a convergence analysis for one-step MAML for a general nonconvex objective. Considering the various source of stochasticity, the learning rates setting is strongly related to various problem-specific parameters, some of which are hard to obtain in reality. Furthermore, Ji et al. (2022) studied the multi-step inner loop MAML's convergence.

## B    THEORETICAL ANALYSIS

### B.1    INTERMEDIATE RESULTS

**Lemma 1** (Lipschitz of the global gradient). *Consider the gradient of the global loss function $F$ as defined in (1). For any $w, u \in \mathbb{R}^d$, we have*

$$\|\nabla F(w) - \nabla F(u)\| \le L \|w - u\|, \tag{9}$$

*where the global objective function's smoothness parameter is*

$$L := \min_{w,u} \{L(w), L(u)\}, \tag{10}$$

$$L(w) := l(1 + \alpha l)^2 + \rho\alpha(1 + \alpha l)\mathbb{E}_i\left[\|\nabla f_i(w)\|\right].$$

*If we further bound the gradient norm as in Assumption 5, we have*

$$L := l(1 + \alpha l)^2 + \rho\alpha G. \tag{11}$$

**Remark 5.** *Notice that by setting $\alpha$ as 0, the dual-loop MAML algorithm degenerate into the conventional multi-task learning problem, and thus $L$ is equal to $l$. In other word, the inner and outer-loop-caused gradient update, resulting in a gradient $F(\boldsymbol{\theta})$ that encompass both the first and second-order gradient of task-specific loss $f_i$. Thus the global gradient $\nabla F(\boldsymbol{\theta})$ becomes more complicated and depends on both the local gradient and hessian smooth parameters, i.e., $l$ and $\rho$.*

*Proof of Lemma 1.* We first recall the definition:

$$\nabla F(w) = \mathbb{E}_i\left[\nabla F_i(w)\right]$$
$$\nabla F_i(w) = \left(\mathbf{I} - \alpha\nabla^2 f_i(w)\right) \cdot f_i\left(w - \alpha\nabla f_i(w)\right).$$

Then we can show that

$$\|\nabla F(w) - \nabla F(u)\| \le \mathbb{E}_i\left[\|\nabla F_i(w) - \nabla F_i(u)\|\right]$$
$$\le \mathbb{E}_i\Big[\underbrace{\|\nabla f_i(w - \alpha\nabla f_i(w)) - \nabla f_i(u - \alpha\nabla f_i(u))\|}_{(i)}$$

$$+ \alpha \underbrace{\left\| \nabla^2 f_i(w) \nabla f_i(w - \alpha \nabla f_i(w)) - \nabla^2 f_i(u) \nabla f_i(u - \alpha \nabla f_i(u)) \right\|}_{(ii)} \bigg]. \quad (12)$$

The first term (i) can be bounded as

$$\|\nabla f_i(w - \alpha \nabla f_i(w)) - \nabla f_i(u - \alpha \nabla f_i(u))\| \leq l \|w - u + \alpha (\nabla f_i(w) - \nabla f_i(u))\|$$
$$\leq l(1 + \alpha l) \|w - u\|. \quad (13)$$

And the second term (ii)

$$\left\| \nabla^2 f_i(w) \nabla f_i(w - \alpha \nabla f_i(w)) - \nabla^2 f_i(u) \nabla f_i(u - \alpha \nabla f_i(u)) \right\|$$
$$= \left\| \nabla^2 f_i(w) \nabla f_i(w - \alpha \nabla f_i(w)) - \nabla^2 f_i(w) \nabla f_i(u - \alpha \nabla f_i(u)) \right.$$
$$\left. + \nabla^2 f_i(w) \nabla f_i(u - \alpha \nabla f_i(u)) - \nabla^2 f_i(u) \nabla f_i(u - \alpha \nabla f_i(u)) \right\|$$
$$\leq \underbrace{\left\| \nabla^2 f_i(w) \right\|}_{\leq l} \cdot \underbrace{\left\| \nabla f_i(w - \alpha \nabla f_i(w)) - \nabla f_i(u - \alpha \nabla f_i(u)) \right\|}_{\leq l(1+\alpha l)\cdot\|w-u\|}$$
$$+ \underbrace{\left\| \nabla^2 f_i(w) - \nabla^2 f_i(u) \right\|}_{\leq \rho\|w-u\|} \cdot \underbrace{\left\| \nabla f_i(u - \alpha \nabla f_i(u)) \right\|}_{G}$$
$$\leq \left( l^2(1 + \alpha l) + \rho G \right) \|w - u\|.$$

The under-braces are due to Assumption 2, 3 and 4, respectively. Based on the bounds of the above two terms, we have

$$\|\nabla F(w) - \nabla F(u)\| \leq \mathbb{E}_i \left[ \left( l(1 + \alpha l) + \alpha \left( l^2(1 + \alpha l) + \rho G \right) \right) \cdot \|w - u\| \right]$$
$$= \left[ l(1 + \alpha l)^2 + \rho \alpha G \right] \cdot \|w - u\|. \quad (14)$$

So that we can define

$$L := l(1 + \alpha l)^2 + \rho \alpha G. \quad (15)$$

Moreover, $\left\| \nabla f_i(u - \alpha \nabla f_i(u)) \right\|$ term can be more tightly bounded by $(1 + \alpha l) \|f_i(u)\|$ using the mean value theorem (Fallah et al., 2020a, Lemma 4.8). That is

$$\|\nabla F(w) - \nabla F(u)\| \leq L \|w - u\|, \quad (16)$$

where

$$L := \min_{w,u} \{L(w), L(u)\}, \quad (17)$$

$$L(w) := l(1 + \alpha l)^2 + \rho \alpha (1 + \alpha l) \mathbb{E}_i \left[ \|\nabla f_i(w)\| \right].$$

$$\square$$

**Lemma 2 (Biased Gradient).** *The outer loop update gradient of MAML, denoted by $\nabla \hat{F}(\boldsymbol{\theta})$, is a biased estimator of $\nabla F(\boldsymbol{\theta})$:*

$$\mathbb{E}_{\mathcal{D}_j^{spt}, \mathcal{D}_j^{qry}} \left[ \nabla f_j \left( \boldsymbol{\theta} - \alpha \nabla f_j(\boldsymbol{\theta}; \mathcal{D}_j^{spt}); \mathcal{D}_j^{qry} \right) \right] = \nabla f_j \left( \boldsymbol{\theta} - \alpha \nabla f_j(\boldsymbol{\theta}) \right) + e_j, \text{ where } \|e_j\| \leq \frac{\alpha l \tilde{\sigma}_G}{\sqrt{\mathcal{D}^{spt}}}.$$

*Proof of Lemma 2.*

$$\mathbb{E}_{\mathcal{D}_j^{spt}, \mathcal{D}_j^{qry}} \left[ \nabla f_j \left( \boldsymbol{\theta} - \alpha \nabla f_j(\boldsymbol{\theta}; \mathcal{D}_j^{spt}); \mathcal{D}_j^{qry} \right) \right]$$
$$= \mathbb{E}_{\mathcal{D}_j^{spt}} \left[ \nabla f_j \left( \boldsymbol{\theta} - \alpha \nabla f_j(\boldsymbol{\theta}; \mathcal{D}_j^{spt}) \right) \right]$$
$$= \mathbb{E}_{\mathcal{D}_j^{spt}} \left[ \nabla f_j \left( \boldsymbol{\theta} - \alpha \nabla f_j(\boldsymbol{\theta}; \mathcal{D}_j^{spt}) \right) - \nabla f_j \left( \boldsymbol{\theta} - \alpha \nabla f_j(\boldsymbol{\theta}) \right) \right] + \nabla f_j \left( \boldsymbol{\theta} - \alpha \nabla f_j(\boldsymbol{\theta}) \right)$$
$$= e_j + \nabla f_j \left( \boldsymbol{\theta} - \alpha \nabla f_j(\boldsymbol{\theta}) \right).$$

$$\|e_j\| \leq \mathbb{E}_{\mathcal{D}_j^{spt}} \left[ \left\| \nabla f_i \left( \boldsymbol{\theta} - \alpha \nabla f_i \left( \boldsymbol{\theta}, \mathcal{D}_j^{spt} \right) \right) - \nabla f_j \left( \boldsymbol{\theta} - \alpha \nabla f_j(\boldsymbol{\theta}) \right) \right\| \right]$$

$$\leq \alpha l \mathbb{E}_{\mathcal{D}_j^{spt}} \left[ \left\| \nabla f_i \left( \boldsymbol{\theta}; \mathcal{D}_j^{spt} \right) - \nabla f_j(\boldsymbol{\theta}) \right\| \right]$$

$$\leq \alpha l \frac{\tilde{\sigma}_G}{\sqrt{D^{spt}}},$$

based on Assumption 2, 4. $\qquad\square$

## B.2 NORMALIZED STOCHASTIC GRADIENT DESCENT WITH MOMENTUM (NSGD-M) UPDATE RULES

The difficulty in analyzing MAML with normalized momentum update is that it is not the case that $\mathbb{E}\left[\boldsymbol{g}_t\right] = \nabla F(\boldsymbol{\theta})$. The inequality comes from two aspects:

- Error of momentum $\boldsymbol{g}_t$ in estimating $\nabla F(\boldsymbol{\theta}_t)$, defined as

$$\boldsymbol{e}_t := \boldsymbol{g}_t - \nabla F(\boldsymbol{\theta}_t). \tag{18}$$

  Notice that $\mathbb{E}\left[\boldsymbol{e}_t\right] \neq 0$, which we will show in proof later.

- Error of $\nabla \hat{F}(\boldsymbol{\theta}_t)$ in estimating $\nabla F(\boldsymbol{\theta})$, defined as

$$\boldsymbol{\epsilon}_t := \nabla \hat{F}(\boldsymbol{\theta}_t) - \nabla F(\boldsymbol{\theta}_t). \tag{19}$$

  From Lemma 2 we know that, $\mathbb{E}\left[\boldsymbol{\epsilon}_t\right] \neq 0$, which means the stochastic gradient of MAML is a **biased estimator** of $\nabla F(\boldsymbol{\theta}_t)$. This error is actually encapsulated in the momentum error $\boldsymbol{e}_t$.

In the following lemma, we are enjoying the convenience brought by the normalized gradient update.

**Lemma 3** (Normalized Gradient Update). *With $\boldsymbol{e}_t := \boldsymbol{g}_t - \nabla F(\boldsymbol{\theta}_t)$, which corresponds to the difference between the updated momentum and the ideal gradient, we have*

$$\frac{1}{T} \sum_{t=0}^{T-1} \mathbb{E}\left[ \|\nabla F(\boldsymbol{\theta}_t)\| \right] \leq \frac{3\Delta}{\beta T} + \frac{8}{T} \sum_{t=0}^{T-1} \mathbb{E}\left[ \|\boldsymbol{e}_t\| \right] + \frac{3\beta L}{2}. \tag{20}$$

*Proof of Lemma 3.* First, from the smoothness condition 1, we have

$$F(\boldsymbol{\theta}_{t+1}) - F(\boldsymbol{\theta}_t) \leq \langle \nabla F(\boldsymbol{\theta}_t), \boldsymbol{\theta}_{t+1} - \boldsymbol{\theta}_t \rangle + \frac{L \|\boldsymbol{\theta}_{t+1} - \boldsymbol{\theta}_t\|^2}{2}$$

$$= -\beta \frac{\langle \nabla F(\boldsymbol{\theta}_t), \boldsymbol{g}_t \rangle}{\|\boldsymbol{g}_t\|} + \frac{L\beta^2}{2}$$

$$\leq -\frac{\beta}{3} \|\nabla F(\boldsymbol{\theta}_t)\| + \frac{8\beta}{3} \|\boldsymbol{e}_t\| + \frac{L\beta^2}{2}, \tag{21}$$

where the last inequality is driven from (Cutkosky & Mehta, 2020, Lemma 2) and can be proved with no further assumptions. Telescoping (21) from $t = 0$ to $t = T - 1$, we got

$$\frac{1}{3} \sum_{t=0}^{T-1} \beta \|\nabla F(\boldsymbol{\theta}_t)\| \leq \Delta + \frac{8}{3} \sum_{t=0}^{T-1} \beta \|\boldsymbol{e}_t\| + \sum_{t=0}^{T-1} \frac{L\beta^2}{2},$$

and thus

$$\frac{1}{T} \sum_{t=0}^{T-1} \|\nabla F(\boldsymbol{\theta}_t)\| \leq \frac{3\Delta}{\beta T} + \frac{8}{T} \sum_{t=0}^{T-1} \|\boldsymbol{e}_t\| + \frac{3\beta L}{2}.$$

$$\frac{1}{T} \sum_{t=0}^{T-1} \mathbb{E}\left[ \|\nabla F(\boldsymbol{\theta}_t)\| \right] \leq \frac{3\Delta}{\beta T} + \frac{8}{T} \sum_{t=0}^{T-1} \mathbb{E}\left[ \|\boldsymbol{e}_t\| \right] + \frac{3\beta L}{2}. \tag{22}$$

$\qquad\square$

Our target now becomes bounding the sum of the stochastic gradients, i.e., $\frac{1}{T} \sum_{t=0}^{T-1} \mathbb{E}\left[ \|\boldsymbol{e}_t\| \right]$.

**Lemma 4** (Recursion of Error of the Momentum). *Define $\epsilon_t = \nabla \hat{F}(\boldsymbol{\theta}_t) - \nabla F(\boldsymbol{\theta}_t)$, $S_t = \nabla F(\boldsymbol{\theta}_t) - \nabla F(\boldsymbol{\theta}_{t+1})$. Then we have*

$$\boldsymbol{e}_t = (1-\gamma)^t \boldsymbol{e}_0 + \sum_{\tau=0}^{t-1} \gamma(1-\gamma)^{t-\tau-1} \boldsymbol{\epsilon}_{\tau+1} + \sum_{\tau=0}^{t-1} (1-\gamma)^{t-\tau} S_\tau \qquad (23)$$

*Proof of Lemma 4.* With the stochastic gradient error $\epsilon_t = \nabla \hat{F}(\boldsymbol{\theta}_t) - \nabla F(\boldsymbol{\theta}_t)$, and the global update in each step $S_t = \nabla F(\boldsymbol{\theta}_t) - \nabla F(\boldsymbol{\theta}_{t+1})$, we have

$$\begin{aligned}
\boldsymbol{e}_{t+1} &= \boldsymbol{g}_{t+1} - \nabla F(\boldsymbol{\theta}_{t+1}) \\
&= (1-\gamma)\boldsymbol{g}_t + \gamma \nabla \hat{F}(\boldsymbol{\theta}_{t+1}) - \nabla F(\boldsymbol{\theta}_{t+1}) \\
&= (1-\gamma)\boldsymbol{e}_t + \gamma \boldsymbol{\epsilon}_{t+1} + (1-\gamma)S_t.
\end{aligned}$$

Finally, unrolling the recursion from $t = T-1$ to $t = 0$, we have

$$\boldsymbol{e}_t = (1-\gamma)^t \boldsymbol{e}_0 + \sum_{\tau=0}^{t-1} \gamma(1-\gamma)^{t-\tau-1} \boldsymbol{\epsilon}_{\tau+1} + \sum_{\tau=0}^{t-1} (1-\gamma)^{t-\tau} S_\tau \qquad (24)$$

$\square$

To bound (22), we apparently need to bound the term $\frac{1}{T}\sum_{t=0}^{T-1} \mathbb{E}\left[\|\boldsymbol{e}_t\|\right]$:

**Lemma 5** (Bound of the Momentum's Error). *Based on the Lemma 4, we can further deduce:*

$$\frac{1}{T}\sum_{t=0}^{T-1} \mathbb{E}\left[\|\boldsymbol{e}_t\|\right] < \frac{1}{\gamma T}\mathbb{E}\left[\|\boldsymbol{e}_0\|\right] + \sqrt{\gamma}\sqrt{\mathbb{E}\left[\|\boldsymbol{\epsilon}\|^2\right]} + \|\mathbb{E}\left[\boldsymbol{\epsilon}\right]\| + \frac{\beta L}{\gamma}. \qquad (25)$$

*Proof of Lemma 5.* We first decompose $\mathbb{E}[\|\boldsymbol{e}_t\|]$ based on Lemma 4 as following

$$\mathbb{E}\left[\|\boldsymbol{e}_t\|\right] \leq \mathbb{E}\left[\left\|(1-\gamma)^t \boldsymbol{e}_0\right\|\right] + \mathbb{E}\left[\left\|\sum_{\tau=0}^{t-1} \gamma(1-\gamma)^{t-\tau-1} \boldsymbol{\epsilon}_{\tau+1}\right\|\right] + \mathbb{E}\left[\left\|\sum_{\tau=0}^{t-1}(1-\gamma)^{t-\tau} S_\tau\right\|\right].$$

$$\frac{1}{T}\sum_{t=0}^{T-1} \mathbb{E}\left[\|\boldsymbol{e}_t\|\right] \leq \underbrace{\frac{1}{T}\sum_{t=0}^{T-1} \mathbb{E}\left[\left\|(1-\gamma)^t \boldsymbol{e}_0\right\|\right]}_{(a)} + \underbrace{\frac{1}{T}\sum_{t=0}^{T-1} \mathbb{E}\left[\left\|\sum_{\tau=0}^{t-1} \gamma(1-\gamma)^{t-\tau-1} \boldsymbol{\epsilon}_{\tau+1}\right\|\right]}_{(b)} + \underbrace{\frac{1}{T}\sum_{t=0}^{T-1} \mathbb{E}\left[\left\|\sum_{\tau=0}^{t-1}(1-\gamma)^{t-\tau} S_\tau\right\|\right]}_{(c)}.$$

$$(26)$$

(a):

$$\begin{aligned}
\frac{1}{T}\sum_{t=0}^{T-1} \mathbb{E}\left[\left\|(1-\gamma)^t \boldsymbol{e}_0\right\|\right] &= \frac{1}{T}\sum_{t=0}^{T-1}(1-\gamma)^t \mathbb{E}\left[\|\boldsymbol{e}_0\|\right] \\
&= \frac{1}{T}\frac{1-(1-\gamma)^T}{\gamma}\mathbb{E}\left[\|\boldsymbol{e}_0\|\right] \\
&< \frac{1}{\gamma T}\mathbb{E}\left[\|\boldsymbol{e}_0\|\right].
\end{aligned} \qquad (27)$$

(b):

$$\frac{1}{T}\sum_{t=0}^{T-1} \mathbb{E}\left[\left\|\sum_{\tau=0}^{t-1} \gamma(1-\gamma)^{t-\tau-1} \boldsymbol{\epsilon}_{\tau+1}\right\|\right] \leq \frac{1}{T}\sum_{t=0}^{T-1} \sqrt{\mathbb{E}\left[\left\|\sum_{\tau=0}^{t-1} \gamma(1-\gamma)^{t-\tau-1} \boldsymbol{\epsilon}_{\tau+1}\right\|^2\right]}$$

$$= \frac{1}{T} \sum_{t=0}^{T-1} \sqrt{\underbrace{\sum_{\tau=0}^{t-1} \gamma^2 (1-\gamma)^{2(t-1-\tau)} \mathbb{E}\left[\|\boldsymbol{\epsilon}_{\tau+1}\|^2\right]}_{(b_1)} + \underbrace{\sum_{0\leq\tau_1<\tau_2\leq t-1}^{t-1} 2\gamma^2 (1-\gamma)^{2(t-1)-\tau_1-\tau_2} \mathbb{E}\left[\langle\boldsymbol{\epsilon}_{\tau_1}, \boldsymbol{\epsilon}_{\tau_2}\rangle\right]}_{(b_2)}}.$$

(b1):

$$\sum_{\tau=0}^{t-1} \gamma^2 (1-\gamma)^{2(t-1-\tau)} \mathbb{E}\left[\|\boldsymbol{\epsilon}_{\tau+1}\|^2\right]$$

$$= \gamma^2 \sum_{\tau=0}^{t-1} (1-\gamma)^{2(t-1-\tau)} \mathbb{E}\left[\|\boldsymbol{\epsilon}_{\tau+1}\|^2\right]$$

$$\leq \gamma^2 \frac{1-(1-\gamma)^{2t}}{1-(1-\gamma)^2} \mathbb{E}\left[\|\boldsymbol{\epsilon}\|^2\right]$$

$$= \gamma \frac{1-(1-\gamma)^T}{2\gamma-\gamma^2} \mathbb{E}\left[\|\boldsymbol{\epsilon}\|^2\right]$$

$$< \gamma^2 \frac{1}{2\gamma-\gamma^2} \sigma^2 < \gamma^2 \frac{1}{\gamma} \mathbb{E}\left[\|\boldsymbol{\epsilon}\|^2\right]$$

$$= \gamma \mathbb{E}\left[\|\boldsymbol{\epsilon}\|^2\right]. \tag{28}$$

Notice the special case when $\gamma = 0$ or $\gamma = 1$, $\gamma^2 (1-\gamma)^{2(t-1-\tau)} = 0$.

(b2):

$$\sum_{0\leq\tau_1<\tau_2\leq t-1}^{t-1} 2\gamma^2 (1-\gamma)^{2(t-1)-\tau_1-\tau_2} \mathbb{E}\left[\langle\boldsymbol{\epsilon}_{\tau_1}, \boldsymbol{\epsilon}_{\tau_2}\rangle\right]$$

$$\leq \sum_{0\leq\tau_1<\tau_2\leq t-1}^{t-1} 2\gamma^2 (1-\gamma)^{2(t-1)-\tau_1-\tau_2} \|\mathbb{E}\left[\boldsymbol{\epsilon}\right]\|^2$$

$$= 2\gamma^2 \left(\frac{1}{1-\gamma}\right)^{-2(t-1)} \left(\frac{\left(\frac{1}{1-\gamma}\right)-\left(\frac{1}{1-\gamma}\right)^{2t-1}}{1-\left(\frac{1}{1-\gamma}\right)^2} + \frac{\left(\frac{1}{1-\gamma}\right)^t-\left(\frac{1}{1-\gamma}\right)^{2t-1}}{1-\left(\frac{1}{1-\gamma}\right)}\right) \frac{1}{1-\left(\frac{1}{1-\gamma}\right)} \|\mathbb{E}\left[\boldsymbol{\epsilon}\right]\|^2$$

$$\leq \|\mathbb{E}\left[\boldsymbol{\epsilon}\right]\|^2. \tag{29}$$

Notice that when $\gamma = 0$ or $\gamma = 1$, $2\gamma^2 (1-\gamma)^{2(t-1)-\tau_1-\tau_2} = 0$. Based on the bound of $(b_1)$ and $(b_2)$, we obtain the bound for $(b)$:

$$\frac{1}{T} \sum_{t=0}^{T-1} \mathbb{E}\left[\left\|\sum_{\tau=0}^{t-1} \gamma(1-\gamma)^{t-\tau-1}\boldsymbol{\epsilon}_{\tau+1}\right\|\right] < \sqrt{\gamma}\sqrt{\mathbb{E}\left[\|\boldsymbol{\epsilon}\|^2\right]} + \|\mathbb{E}\left[\boldsymbol{\epsilon}\right]\|. \tag{30}$$

(c):

$$\frac{1}{T} \sum_{t=0}^{T-1} \mathbb{E}\left[\left\|\sum_{\tau=0}^{t-1} (1-\gamma)^{t-\tau} S_\tau\right\|\right] = \frac{1}{T} \sum_{t=0}^{T-1} \mathbb{E}\left[\left\|\frac{(1-\gamma)-(1-\gamma)^{t+1}}{\gamma} L\right\|\right]$$

$$\leq \frac{1}{T} \sum_{t=0}^{T-1} \frac{(1-\gamma)-(1-\gamma)^{t+1}}{\gamma} \beta L = \frac{1}{T} \frac{(1-\gamma)T - \sum_{t=0}^{T-1}(1-\gamma)^{t+1}}{\gamma} \beta L$$

$$= \frac{1}{T} \frac{(1-\gamma)T - \frac{(1-\gamma)-(1-\gamma)^{T+1}}{\gamma}}{\gamma} \beta L = \left(\frac{1-\gamma}{\gamma} - \frac{(1-\gamma)-(1-\gamma)^{T+1}}{\gamma^2 T}\right) \beta L$$

$$< \frac{\beta L}{\gamma}$$

$$(31)$$

based on Lemma 1 and the normalized update rule, along with the basic summing a geometric sequence.

Combing (a), (b) and (c) in (26), we have

$$\frac{1}{T} \sum_{t=0}^{T-1} \mathbb{E}\left[\|e_t\|\right] < \frac{1}{\gamma T} \mathbb{E}\left[\|e_0\|\right] + \sqrt{\gamma} \sqrt{\mathbb{E}\left[\|\boldsymbol{\epsilon}\|^2\right]} + \|\mathbb{E}\left[\boldsymbol{\epsilon}\right]\| + \frac{\beta L}{\gamma}. \tag{32}$$

$\square$

### B.3 CONVERGENCE OF MAML

Following equation equation 25, it remains to bound the terms $\mathbb{E}\left[\|\boldsymbol{\epsilon}\|^2\right]$, $\mathbb{E}\left[\|\boldsymbol{\epsilon}\|\right]$ appearing in equation 25. By definition, the error term $\boldsymbol{\epsilon}_t = \nabla \hat{F}(\boldsymbol{\theta}_t) - \nabla F(\boldsymbol{\theta}_t)$ captures the stochastic deviation of the outer-loop gradient in MAML from its true expectation. As elaborated in Section 2.1, this stochasticity primarily arises from two sources: (i) sampling variability within each task's data (inner-loop sampling), and (ii) randomness in task selection (meta-level or outer-loop sampling). Both sources contribute cumulatively to the bound on $\mathbb{E}\left[\|\boldsymbol{\epsilon}\|^2\right]$.

In the following analysis, we consider an arbitrary iteration $t$ and omit the subscript for notational simplicity. Based on the definitions (1) and (2), we have $\mathbb{E}\left[\|\boldsymbol{\epsilon}\|^2\right]$:

$$\mathbb{E}\left[\|\boldsymbol{\epsilon}\|^2\right] = \mathbb{E}\left[\left\|\nabla \hat{F}(\boldsymbol{\theta}) - \nabla F(\boldsymbol{\theta})\right\|^2\right] \tag{33}$$

**Lemma 6.** *Bound on* $\mathbb{E}\left[\|\boldsymbol{\epsilon}\|^2\right]$ *Based on the Assumptions, we have*

$$\mathbb{E}\left[\|\boldsymbol{\epsilon}\|^2\right] \leq \frac{10}{J}\left[\frac{G^2\alpha^2\tilde{\sigma}_H^2}{D^{spt}} + \frac{(1+\alpha l)^2\alpha^2 l^2\tilde{\sigma}_G^2}{D^{spt}} + \frac{\alpha^4\tilde{\sigma}_H^2\tilde{\sigma}_G^2}{(D^{spt})^2} + \frac{(1+\alpha l)^2\tilde{\sigma}_G^2}{D^{qry}} + \frac{\alpha^2\tilde{\sigma}_H^2\tilde{\sigma}_G^2}{D^{spt}D^{qry}}\right]$$

$$+ \frac{6}{J}\left[\alpha^2\sigma_H^2 G^2 + 2(1+\alpha l)^2\left(1+(\alpha l)^2\right)\sigma_G^2 + 2\alpha^2\left(1+(\alpha l)^2\right)\sigma_G^2\sigma_H^2\right]. \tag{34}$$

Before the formal proof of Lemma 6, recall the definitions (1) and (2):

$$\nabla F(\boldsymbol{\theta}) = \frac{1}{I} \sum_{i \in \mathcal{I}} \left[\left(\mathbf{I} - \alpha\nabla^2 f_i(\boldsymbol{\theta})\right) \cdot \nabla f_i\left(\theta - \alpha\nabla f_i(\boldsymbol{\theta})\right)\right].$$

$$\nabla \hat{F}(\boldsymbol{\theta}) = \frac{1}{J} \sum_{j \in \mathcal{I}_t} \left[\left(\mathbf{I} - \alpha\nabla^2 f_j\left(\theta; \mathcal{D}_j^{spt}\right)\right) \cdot \nabla f_j\left(\theta - \alpha\nabla f_j\left(\theta; \mathcal{D}_j^{spt}\right); \mathcal{D}_j^{qry}\right)\right].$$

Form these two gradients, it is hard to directly bound the stochasticity. Thereby we define an intermediate gradient, denoted by $\nabla \tilde{F}(\boldsymbol{\theta})$, which omitted the randomness brought by data sampling compared with $\nabla \hat{F}(\boldsymbol{\theta})$ but still remains random in tasks sampling compared with $\nabla F(\boldsymbol{\theta})$:

$$\nabla \tilde{F}(\boldsymbol{\theta}) = \frac{1}{J} \sum_{j \in \mathcal{I}_t} \left[\left(\mathbf{I} - \alpha\nabla^2 f_j(\boldsymbol{\theta})\right) \cdot \nabla f_j\left(\theta - \alpha\nabla f_j(\boldsymbol{\theta})\right)\right]. \tag{35}$$

Consequently, we may decompose (33) as follows

$$\mathbb{E}\left[\left\|\nabla \hat{F}(\boldsymbol{\theta}) - \nabla F(\boldsymbol{\theta})\right\|^2\right]$$

$$= \mathbb{E}\left[\left\|\nabla \hat{F}(\boldsymbol{\theta}) - \nabla \tilde{F}(\boldsymbol{\theta}) + \nabla \tilde{F}(\boldsymbol{\theta}) - \nabla F(\boldsymbol{\theta})\right\|^2\right]$$

$$\leq 2\underbrace{\mathbb{E}\left[\left\|\nabla\hat{F}(\boldsymbol{\theta})-\nabla\tilde{F}(\boldsymbol{\theta})\right\|^2\right]}_{(d)}+2\underbrace{\mathbb{E}\left[\left\|\nabla\tilde{F}(\boldsymbol{\theta})-\nabla F(\boldsymbol{\theta})\right\|^2\right]}_{(e)}. \tag{36}$$

**Lemma 7.** *Bound on (d):* $\mathbb{E}\left[\left\|\nabla\hat{F}(\boldsymbol{\theta})-\nabla\tilde{F}(\boldsymbol{\theta})\right\|^2\right]$.

$$\mathbb{E}\left[\left\|\nabla\hat{F}(\boldsymbol{\theta})-\nabla\tilde{F}(\boldsymbol{\theta})\right\|^2\right] \leq \frac{5}{J}\left[\frac{G^2\alpha^2\tilde{\sigma}_H^2}{D^{spt}}+\frac{(1+\alpha l)^2\alpha^2 l^2\tilde{\sigma}_G^2}{D^{spt}}+\frac{\alpha^4 l^2\tilde{\sigma}_H^2\tilde{\sigma}_G^2}{\left(D^{spt}\right)^2}+\frac{(1+\alpha l)^2\tilde{\sigma}_G^2}{D^{qry}}+\frac{\alpha^2\tilde{\sigma}_H^2\tilde{\sigma}_G^2}{D^{spt}D^{qry}}\right].$$
$$\tag{37}$$

*Proof.* Proof of Lemma 7. **For term (d)**, by definition we have

$$\mathbb{E}\left[\left\|\nabla\hat{F}(\boldsymbol{\theta})-\nabla\tilde{F}(\boldsymbol{\theta})\right\|^2\right]$$

$$= \mathbb{E}\left[\left\|\frac{1}{J}\sum_{j\in\mathcal{I}_t}\left[(\mathbf{I}-\alpha\nabla^2 f_j(\theta;\mathcal{D}_j^{spt}))\cdot\nabla f_j\left(\theta-\alpha\nabla f_j(\theta;\mathcal{D}_j^{spt});\mathcal{D}_j^{qry}\right)\right]-\frac{1}{J}\sum_{j\in\mathcal{I}_t}\left[(\mathbf{I}-\alpha\nabla^2 f_j(\boldsymbol{\theta}))\cdot\nabla f_j\left(\theta-\alpha\nabla f_j(\boldsymbol{\theta})\right)\right]\right\|^2\right]$$

$$\leq \frac{1}{J}\mathbb{E}\left[\left\|(\mathbf{I}-\alpha\nabla^2 f_j(\theta;\mathcal{D}_j^{spt}))\cdot\nabla f_j\left(\theta-\alpha\nabla f_j(\theta;\mathcal{D}_j^{spt});\mathcal{D}_j^{qry}\right)-(\mathbf{I}-\alpha\nabla^2 f_j(\boldsymbol{\theta}))\cdot\nabla f_j\left(\theta-\alpha\nabla f_j(\boldsymbol{\theta})\right)\right\|^2\right]$$

$$= \frac{1}{J}\mathbb{E}\left[\left\|\left((\mathbf{I}-\alpha\nabla^2 f_j(\boldsymbol{\theta}))+e_{H,j}^{spt}\right)\cdot\left((\nabla f_j\left(\theta-\alpha\nabla f_j(\boldsymbol{\theta})\right)+e_{G,j}^{spt})+e_{G,j}^{qry}\right)-(\mathbf{I}-\alpha\nabla^2 f_j(\boldsymbol{\theta}))\cdot\nabla f_j\left(\theta-\alpha\nabla f_j(\boldsymbol{\theta})\right)\right\|^2\right],$$
$$\tag{38}$$

where the inequality is due to the convexity of norm operation. The colored error terms are defined as following. First, the hessian error caused by the support samples' randomness is $e_{H,j}^{spt}$, expressed as

$$e_{H,j}^{spt} = \left(\mathbf{I}-\alpha\nabla^2 f_j\left(\theta;\mathcal{D}_j^{spt}\right)\right)-\left(\mathbf{I}-\alpha\nabla^2 f_j\left(\theta\right)\right)=\alpha\left(\nabla^2 f_j(\boldsymbol{\theta})-\nabla^2 f_j\left(\theta;\mathcal{D}_j^{spt}\right)\right). \tag{39}$$

From Assumption 4, we have

$$\mathbb{E}\left[\left\|e_{H,j}^{spt}\right\|\right]\leq\frac{\alpha\tilde{\sigma}_H}{\sqrt{D^{spt}}}. \tag{40}$$

Likely, the gradient error $e_{G,j}^{spt}$ caused by the support sample:

$$e_{G,j}^{spt}=\nabla f_j\left(\theta-\alpha\nabla f_j\left(\theta;\mathcal{D}_j^{spt}\right)\right)-\nabla f_j\left(\theta-\alpha\nabla f_j\left(\theta\right)\right), \tag{41}$$

which is bounded as

$$\mathbb{E}\left[\left\|e_{G,j}^{spt}\right\|\right] = \mathbb{E}\left[\left\|\nabla f_j\left(\theta-\alpha\nabla f_j\left(\theta;\mathcal{D}_j^{spt}\right)\right)-\nabla f_j\left(\theta-\alpha\nabla f_j\left(\theta\right)\right)\right\|\right]$$

$$\leq l\mathbb{E}\left[\left\|\alpha\nabla f_j\left(\theta;\mathcal{D}_j^{spt}\right)-\alpha\nabla f_j(\boldsymbol{\theta})\right\|\right]$$

$$\leq \frac{\alpha l\tilde{\sigma}_G}{\sqrt{D^{spt}}}, \tag{42}$$

where the first inequality comes from Assumption 2 and the second is from Assumption 4.

Besides, the query sample-caused gradient error $e_{G,j}^{qry}$ is defined as

$$e_{G,j}^{qry}=\nabla f_j\left(\theta-\alpha\nabla f_j\left(\theta;\mathcal{D}_j^{spt}\right);\mathcal{D}_j^{qry}\right)-\nabla f_j\left(\theta-\alpha\nabla f_j\left(\theta;\mathcal{D}_j^{spt}\right)\right), \tag{43}$$

and according to Assumption 4, we have the error bound

$$\mathbb{E}\left[\left\|e_{G,j}^{qry}\right\|\right]\leq\frac{\tilde{\sigma}_G}{\sqrt{D^{qry}}}. \tag{44}$$

Based on these, we further rewrite (38) as

$$\frac{1}{J}\mathbb{E}\left[\left\|e_{H,j}^{spt}\nabla f_j\left(\theta-\alpha\nabla f_j(\boldsymbol{\theta})\right)+\left(\mathbf{I}-\alpha\nabla^2 f_j(\boldsymbol{\theta})\right)e_{G,j}^{spt}+e_{H,j}^{spt}e_{G,j}^{spt}+\left(\mathbf{I}-\alpha\nabla^2 f_j(\boldsymbol{\theta})\right)e_{G,j}^{qry}+e_{H,j}^{spt}e_{G,j}^{qry}\right\|^2\right]$$

$$\leq \frac{1}{J}\mathbb{E}\left[\left(\left\|e_{H,j}^{spt}\nabla f_j\left(\theta - \alpha\nabla f_j(\boldsymbol{\theta})\right)\right\| + \left\|\left(\mathbf{I} - \alpha\nabla^2 f_j(\boldsymbol{\theta})\right)e_{G,j}^{spt}\right\| + \left\|e_{H,j}^{spt}e_{G,j}^{spt}\right\| + \left\|\left(\mathbf{I} - \alpha\nabla^2 f_j(\boldsymbol{\theta})\right)e_{G,j}^{qry}\right\| + \left\|e_{H,j}^{spt}e_{G,j}^{qry}\right\|\right)^2\right]$$

$$\leq \frac{5}{J}\left[\mathbb{E}\left[\left\|e_{H,j}^{spt}\nabla f_j\left(\theta - \alpha\nabla f_j(\boldsymbol{\theta})\right)\right\|^2\right] + \mathbb{E}\left[\left\|\left(\mathbf{I} - \alpha\nabla^2 f_j(\boldsymbol{\theta})\right)e_{G,j}^{spt}\right\|^2\right]\right.$$

$$\left. + \mathbb{E}\left[\left\|e_{H,j}^{spt}e_{G,j}^{spt}\right\|^2\right] + \mathbb{E}\left[\left\|\left(\mathbf{I} - \alpha\nabla^2 f_j(\boldsymbol{\theta})\right)e_{G,j}^{qry}\right\|^2\right] + \mathbb{E}\left[\left\|e_{H,j}^{spt}e_{G,j}^{qry}\right\|^2\right]\right]$$

$$\leq \frac{5}{J}\left[\frac{G^2\alpha^2\tilde{\sigma}_H^2}{D^{spt}} + \frac{(1+\alpha l)^2\alpha^2 l^2\tilde{\sigma}_G^2}{D^{spt}} + \frac{\alpha^4 l^2\tilde{\sigma}_H^2\tilde{\sigma}_G^2}{\left(D^{spt}\right)^2} + \frac{(1+\alpha l)^2\tilde{\sigma}_G^2}{D^{qry}} + \frac{\alpha^2\tilde{\sigma}_H^2\tilde{\sigma}_G^2}{D^{spt}D^{qry}}\right]. \tag{45}$$

$\square$

**Lemma 8.** *Bound on (e):* $\mathbb{E}\left[\left\|\nabla\tilde{F}(\boldsymbol{\theta}) - \nabla F(\boldsymbol{\theta})\right\|^2\right]$.

$$\mathbb{E}\left[\left\|\nabla\tilde{F}(\boldsymbol{\theta}) - \nabla F(\boldsymbol{\theta})\right\|^2\right] \leq \frac{3}{J}\left[\alpha^2\sigma_H^2 G^2 + 2(1+\alpha l)^2\left(1+(\alpha l)^2\right)\sigma_G^2 + 2\alpha^2\left(1+(\alpha l)^2\right)\sigma_G^2\sigma_H^2\right]. \tag{46}$$

*Proof.* Proof of Lemma 8

For the term (e):

$$\mathbb{E}\left[\left\|\nabla\tilde{F}(\boldsymbol{\theta}) - \nabla F(\boldsymbol{\theta})\right\|^2\right]$$

$$= \mathbb{E}\left[\left\|\frac{1}{J}\sum_{j\in\mathcal{I}_t}\nabla F_j(\boldsymbol{\theta}) - \nabla F(\boldsymbol{\theta})\right\|^2\right]$$

$$\leq \frac{1}{J}\mathbb{E}\left[\left\|\nabla F_j(\boldsymbol{\theta}) - \nabla F(\boldsymbol{\theta})\right\|^2\right]$$

$$= \frac{1}{J}\mathbb{E}\left[\left\|\left(\mathbf{I} - \alpha\nabla^2 f_j(\boldsymbol{\theta})\right)\nabla f_j\left(\theta - \alpha\nabla f_j(\boldsymbol{\theta})\right) - \left(\mathbf{I} - \alpha\nabla^2 f(\boldsymbol{\theta})\right)\nabla f\left(\theta - \alpha\nabla f(\boldsymbol{\theta})\right)\right\|^2\right]$$

$$= \frac{1}{J}\mathbb{E}\left[\left\|\left(\left(\mathbf{I} - \alpha\nabla^2 f(\boldsymbol{\theta})\right) + e_{H,j}\right)\left(\nabla f\left(\theta - \alpha\nabla f(\boldsymbol{\theta})\right) + e_{G,j}\right) - \left(\mathbf{I} - \alpha\nabla^2 f(\boldsymbol{\theta})\right)\nabla f\left(\theta - \alpha\nabla f(\boldsymbol{\theta})\right)\right\|^2\right] \tag{47}$$

The error terms resulted by the randomness of tasks are defined as follow. First, $e_{H,j}$ is the random task-inducing hessian error, defined as

$$e_{H,j} = \left(\mathbf{I} - \alpha\nabla^2 f_j(\boldsymbol{\theta})\right) - \left(\mathbf{I} - \alpha\nabla^2 f(\boldsymbol{\theta})\right) = \alpha\left(\nabla^2 f_j(\boldsymbol{\theta}) - \nabla^2 f(\boldsymbol{\theta})\right), \tag{48}$$

and whose expected norm can be bounded based on the assumption

$$\mathbb{E}\left[\|e_{H,j}\|\right] \leq \alpha\sigma_H. \tag{49}$$

Then the the random task-inducing gradient error $e_{G,j}$ is

$$e_{G,j} = \nabla f_j\left(\theta - \alpha\nabla f_j(\boldsymbol{\theta})\right) - \nabla f\left(\theta - \alpha\nabla f(\boldsymbol{\theta})\right), \tag{50}$$

with the expected norm bounded as

$$\mathbb{E}\left[\|e_{G,j}\|\right] \leq \sqrt{2 + 2(\alpha\ell)^2}\sigma_G, \tag{51}$$

based on Assumption 3, since

$$e_{G,j} = \nabla f_j\left(\theta - \alpha\nabla f_j(\boldsymbol{\theta})\right) - \nabla f\left(\theta - \alpha\nabla f(\boldsymbol{\theta})\right)$$
$$= \left[\nabla f_j\left(\theta - \alpha\nabla f_j(\boldsymbol{\theta})\right) - \nabla f_j\left(\theta - \alpha\nabla f(\boldsymbol{\theta})\right)\right] + \left[\nabla f_j\left(\theta - \alpha\nabla f(\boldsymbol{\theta})\right) - \nabla f\left(\theta - \alpha\nabla f(\boldsymbol{\theta})\right)\right].$$

Therefore,

$$\mathbb{E}\left[\|e_{G,j}\|\right] \leq \sqrt{\mathbb{E}\left[\|e_{G,j}\|^2\right]}$$

$$\leq \sqrt{2\mathbb{E}\left[\left\|\nabla f_j\left(\theta - \alpha\nabla f_j(\boldsymbol{\theta})\right) - \nabla f_j\left(\theta - \alpha\nabla f(\boldsymbol{\theta})\right)\right\|^2\right] + 2\mathbb{E}\left[\left\|\nabla f_j\left(\theta - \alpha\nabla f(\boldsymbol{\theta})\right) - \nabla f\left(\theta - \alpha\nabla f(\boldsymbol{\theta})\right)\right\|^2\right]}$$

$$\leq \sqrt{2\alpha^2\ell^2\sigma_G^2 + 2\sigma_G^2}$$

$$= \sqrt{2\left(\alpha^2\ell^2 + 1\right)}\sigma_G,$$

where the three inequalities is based on the convexity of $l_2$ norm, the triangle inequality and Assumption 3, respectively.

With (49) and (52), we can further write (47) as

$$\frac{1}{J}\mathbb{E}\left[\left\|\left(\left(\mathbf{I} - \alpha\nabla^2 f(\boldsymbol{\theta})\right) + e_{H,j}\right)\left(\nabla f_j\left(\theta - \alpha\nabla f_j(\boldsymbol{\theta})\right) + e_{G,j}\right) - \left(\mathbf{I} - \alpha\nabla^2 f(\boldsymbol{\theta})\right)\nabla f\left(\theta - \alpha\nabla f(\boldsymbol{\theta})\right)\right\|^2\right]$$

$$= \frac{1}{J}\mathbb{E}\left[\left\|e_{H,j}\nabla f_j\left(\theta - \alpha\nabla f_j(\boldsymbol{\theta})\right) + \left(\mathbf{I} - \alpha\nabla^2 f(\boldsymbol{\theta})\right)e_{G,j} + e_{H,j}e_{G,j}\right\|^2\right]$$

$$\leq \frac{1}{J}\mathbb{E}\left[\left(\left\|e_{H,j}\nabla f_j\left(\theta - \alpha\nabla f_j(\boldsymbol{\theta})\right)\right\| + \left\|\left(\mathbf{I} - \alpha\nabla^2 f(\boldsymbol{\theta})\right)e_{G,j}\right\| + \left\|e_{H,j}e_{G,j}\right\|\right)^2\right]$$

$$\leq \frac{3}{J}\left[\mathbb{E}\left[\left\|e_{H,j}\nabla f_j\left(\theta - \alpha\nabla f_j(\boldsymbol{\theta})\right)\right\|^2\right] + \mathbb{E}\left[\left\|\left(\mathbf{I} - \alpha\nabla^2 f(\boldsymbol{\theta})\right)e_{G,j}\right\|^2\right] + \mathbb{E}\left[\left\|e_{H,j}e_{G,j}\right\|^2\right]\right]$$

$$\leq \frac{3}{J}\left[\alpha^2\sigma_H^2 G^2 + 2(1 + \alpha l)^2\left(1 + (\alpha l)^2\right)\sigma_G^2 + 2\alpha^2\left(1 + (\alpha l)^2\right)\sigma_G^2\sigma_H^2\right]. \tag{52}$$

$\square$

*Proof.* Proof for Lemma 6.

We can assemble (36) using Lemma 7 and 8 as

$$\mathbb{E}\left[\|\boldsymbol{\epsilon}\|^2\right] = \mathbb{E}\left[\left\|\nabla\hat{F}(\boldsymbol{\theta}) - \nabla F(\boldsymbol{\theta})\right\|^2\right]$$

$$\leq 2\underbrace{\mathbb{E}\left[\left\|\nabla\hat{F}(\boldsymbol{\theta}) - \nabla\tilde{F}(\boldsymbol{\theta})\right\|^2\right]}_{(d)} + 2\underbrace{\mathbb{E}\left[\left\|\nabla\tilde{F}(\boldsymbol{\theta}) - \nabla F(\boldsymbol{\theta})\right\|^2\right]}_{(e)}$$

$$\leq \underbrace{\frac{10}{J}\left[\frac{G^2\alpha^2\tilde{\sigma}_H^2}{D^{spt}} + \frac{(1 + \alpha l)^2\alpha^2 l^2\tilde{\sigma}_G^2}{D^{spt}} + \frac{\alpha^4\tilde{\sigma}_H^2\tilde{\sigma}_G^2}{(D^{spt})^2} + \frac{(1 + \alpha l)^2\tilde{\sigma}_G^2}{D^{qry}} + \frac{\alpha^2\tilde{\sigma}_H^2\tilde{\sigma}_G^2}{D^{spt}D^{qry}}\right]}_{(d)}$$

$$+ \underbrace{\frac{6}{J}\left[\alpha^2\sigma_H^2 G^2 + 2(1 + \alpha l)^2\left(1 + (\alpha l)^2\right)\sigma_G^2 + 2\alpha^2\left(1 + (\alpha l)^2\right)\sigma_G^2\sigma_H^2\right]}_{(e)} \tag{53}$$

$\square$

**Lemma 9.** ***Bound on*** $\|\mathbb{E}\left[\boldsymbol{\epsilon}\right]\|$. *The norm of the expected error is bounded as*

$$\|\mathbb{E}\left[\boldsymbol{\epsilon}\right]\| \leq \sqrt{(1 + \alpha l)^2(\alpha l)^2\frac{\tilde{\sigma}_G^2}{D^{spt}}}. \tag{54}$$

*Proof.* Proof of Lemma 9

By definition, we have

$$\|\mathbb{E}\left[\boldsymbol{\epsilon}\right]\| = \left\|\mathbb{E}\left[\nabla\hat{F}(\boldsymbol{\theta})\right] - \nabla F(\boldsymbol{\theta})\right\|. \tag{55}$$

$$\nabla\hat{F}(\boldsymbol{\theta}) = \frac{1}{J}\sum_{j\in\mathcal{I}_t}\left[\left(\mathbf{I} - \alpha\nabla^2 f_j(\theta; \mathcal{D}_j^{spt})\right)\cdot\nabla f_j\left(\theta - \alpha\nabla f_j(\theta; \mathcal{D}_j^{spt}); \mathcal{D}_j^{qry}\right)\right] \tag{56}$$

$$\mathbb{E}\left[\nabla\hat{F}(\boldsymbol{\theta})\right] = \mathbb{E}_j\left[\mathbb{E}_{D_j^{spt}}\left[\left(\mathbf{I} - \alpha\nabla^2 f_j(\theta; \mathcal{D}_j^{spt})\right)\right]\cdot\mathbb{E}_{\mathcal{D}_j^{qry}, \mathcal{D}_j^{spt}}\left[\nabla f_j(\theta - \alpha\nabla f_j(\theta; \mathcal{D}_j^{spt}); \mathcal{D}_j^{qry})\right]\right]$$

$$= \mathbb{E}_j \left[ (\mathbf{I} - \alpha\nabla^2 f_j(\boldsymbol{\theta})) \cdot (\nabla f_j (\theta - \alpha\nabla f_j(\boldsymbol{\theta})) + \boldsymbol{e}_j) \right], \tag{57}$$

where $\boldsymbol{e}_j$ is given by

$$\boldsymbol{e}_j := \mathbb{E}_{\mathcal{D}_j^{qry}, \mathcal{D}_j^{spt}} \left[ \nabla f_j(\theta - \alpha\nabla f_j(\theta; \mathcal{D}_j^{spt}); \mathcal{D}_j^{qry}) \right] - \nabla f_j (\theta - \alpha\nabla f_j(\boldsymbol{\theta})). \tag{58}$$

By simplifying the right hand side of (57), we obtain

$$\mathbb{E} \left[ \nabla\hat{F}(\boldsymbol{\theta}) \right] = \mathbb{E}_j \left[ (\mathbf{I} - \alpha\nabla^2 f_j(\boldsymbol{\theta})) \cdot \nabla f_j (\theta - \alpha\nabla f_j(\boldsymbol{\theta})) + (\mathbf{I} - \alpha\nabla^2 f_j(\boldsymbol{\theta})) \boldsymbol{e}_j \right]$$

$$= \mathbb{E}_j \left[ \nabla F_j(\boldsymbol{\theta}) + (\mathbf{I} - \alpha\nabla^2 f_j(\boldsymbol{\theta})) \cdot \boldsymbol{e}_j \right]$$

$$= \nabla F(\boldsymbol{\theta}) + \mathbb{E} [\boldsymbol{\epsilon}] \tag{59}$$

and the residue is given by $\mathbb{E} [\boldsymbol{\epsilon}] = \mathbb{E}_j [(\mathbf{I} - \alpha\nabla f_j(\boldsymbol{\theta})) \cdot \boldsymbol{e}_j]$. Not that the second equality in (59) is due to the definition $F_j(\boldsymbol{\theta}) := f_j (\theta - \alpha\nabla f_j(\boldsymbol{\theta}))$. Next, we derive an upper bound on the norm of the expected residue $\mathbb{E} [\boldsymbol{\epsilon}]$ as

$$\|\mathbb{E} [\boldsymbol{\epsilon}]\| \le \mathbb{E}_j \left[ \left\| \mathbf{I} - \alpha\nabla^2 f_j(\boldsymbol{\theta}) \right\| \cdot \|\boldsymbol{e}_j\| \right]$$

$$\le (1 + \alpha l)\alpha l \frac{\tilde{\sigma}_G}{\sqrt{D^{spt}}},$$

where the second inequality follows the same proof technique as (42). $\qquad\square$

Based on Lemma 6 and 9, finally we can write (25) as

$$\frac{1}{T} \sum_{t=0}^{T-1} \mathbb{E} [\|\boldsymbol{e}_t\|] < \frac{1}{\gamma T}\mathbb{E} [\|\boldsymbol{e}_0\|] + \sqrt{\gamma}\sqrt{\mathbb{E} \left[ \|\boldsymbol{\epsilon}\|^2 \right]} + \|\mathbb{E} [\boldsymbol{\epsilon}]\| + \frac{\beta L}{\gamma}$$

$$\le \left( \frac{1}{\gamma T} + \sqrt{\gamma} \right) \sqrt{\mathbb{E} \left[ \|\boldsymbol{\epsilon}\|^2 \right]} + \|\mathbb{E} [\boldsymbol{\epsilon}]\| + \frac{\beta L}{\gamma}$$

$$\le \left( \frac{1}{\gamma T} + \sqrt{\gamma} \right) \left[ \sqrt{\frac{10}{J}} \left( \frac{G\alpha\tilde{\sigma}_H}{\sqrt{D^{spt}}} + \frac{(1 + \alpha l)\alpha l\tilde{\sigma}_G}{\sqrt{D^{spt}}} + \frac{\alpha^2\tilde{\sigma}_H\tilde{\sigma}_G}{D^{spt}} + \frac{(1 + \alpha l)\tilde{\sigma}_G}{\sqrt{D^{qry}}} + \frac{\alpha\tilde{\sigma}_H\tilde{\sigma}_G}{\sqrt{D^{spt}D^{qry}}} \right) \right.$$

$$\left. + \sqrt{\frac{6}{J}} \left( \alpha\sigma_H G + (1 + \alpha l)\sqrt{2 + 2(\alpha l)^2}\sigma_G + \alpha\sqrt{2 + 2(\alpha l)^2}\sigma_G\sigma_H \right) \right] + (1 + \alpha l)\alpha l\frac{\tilde{\sigma}_G}{\sqrt{D}^{spt}} + \frac{\beta L}{\gamma}$$

Consequently we have the RHS of Eq. (22) bounded as

$$\frac{1}{T} \sum_{t=0}^{T-1} \mathbb{E} [\|\nabla F(\boldsymbol{\theta}_t)\|] \le \frac{3\Delta}{\beta T} + \frac{8}{T} \sum_{t=0}^{T-1} \mathbb{E} [\|\boldsymbol{e}_t\|] + \frac{3\beta L}{2}$$

$$\le \frac{3\Delta}{\beta T} + \left( \frac{8}{\gamma T} + 8\sqrt{\gamma} \right) \left[ \sqrt{\frac{10}{J}} \left( \frac{G\alpha\tilde{\sigma}_H}{\sqrt{D^{spt}}} + \frac{(1 + \alpha l)\alpha l\tilde{\sigma}_G}{\sqrt{D^{spt}}} + \frac{\alpha^2\tilde{\sigma}_H\tilde{\sigma}_G}{D^{spt}} + \frac{(1 + \alpha l)\tilde{\sigma}_G}{\sqrt{D^{qry}}} + \frac{\alpha\tilde{\sigma}_H\tilde{\sigma}_G}{\sqrt{D^{spt}D^{qry}}} \right) \right.$$

$$\left. + \sqrt{\frac{6}{J}} \left( \alpha\sigma_H G + (1 + \alpha l)\sqrt{2 + 2(\alpha l)^2}\sigma_G + \alpha\sqrt{2 + 2(\alpha l)^2}\sigma_G\sigma_H \right) \right] + 8(1 + \alpha l)\alpha l\frac{\tilde{\sigma}_G}{\sqrt{D^{spt}}} + \frac{8\beta l}{\gamma} + \frac{3\beta L}{2}$$

$$\simeq \frac{3\Delta}{\beta T} + \left( \frac{8}{\gamma T} + 8\sqrt{\gamma} \right) \left( \sqrt{\frac{10}{J}}\frac{\tilde{\sigma}_G}{\sqrt{D^{qry}}} + 2\sqrt{\frac{3}{J}}\sigma_G \right) + 8\alpha l\frac{\tilde{\sigma}_G}{\sqrt{D^{spt}}} + \frac{8\beta l}{\gamma}.$$

We set $\alpha = \frac{\sqrt[4]{D^{spt}}}{T^{\frac{1}{4}}}$, $\beta = \frac{D^{qry} \times J}{T^{\frac{3}{4}}}$ and $\gamma = \frac{D^{qry} \times J}{T^{\frac{1}{2}}}$, then we have

$$\frac{1}{T} \sum_{t=0}^{T-1} \mathbb{E} [\|\nabla F(\boldsymbol{\theta}_t)\|] \le \frac{3\Delta}{\beta T} + \frac{8}{T} \sum_{t=0}^{T-1} \mathbb{E} [\|\boldsymbol{e}_t\|] + \frac{3\beta L}{2}$$

$$\le \frac{3\Delta}{\beta T} + \left( \frac{8}{\gamma T} + 8\sqrt{\gamma} \right) \left[ \sqrt{\frac{10}{J}} \left( \frac{G\alpha\tilde{\sigma}_H}{\sqrt{D^{spt}}} + \frac{(1 + \alpha l)\alpha l\tilde{\sigma}_G}{\sqrt{D^{spt}}} + \frac{\alpha^2\tilde{\sigma}_H\tilde{\sigma}_G}{D^{spt}} + \frac{(1 + \alpha l)\tilde{\sigma}_G}{\sqrt{D^{qry}}} + \frac{\alpha\tilde{\sigma}_H\tilde{\sigma}_G}{\sqrt{D^{spt}D^{qry}}} \right) \right.$$

$$+\sqrt{\frac{6}{J}}\left(\alpha\sigma_H G + (1+\alpha l)\sqrt{2+2(\alpha l)^2}\sigma_G + \alpha\sqrt{2+2(\alpha l)^2}\sigma_G\sigma_H\right)\Big] + 8(1+\alpha l)\alpha l\frac{\tilde{\sigma}_G}{\sqrt{D^{spt}}} + \frac{8\beta l}{\gamma} + \frac{3\beta L}{2}$$

$$\simeq \frac{8\alpha^2 l^2\tilde{\sigma}_G}{\sqrt{D^{spt}}} + 8\sqrt{10}\left(\frac{\tilde{\sigma}_G}{\gamma T\sqrt{D^{qry}J}} + \frac{\sqrt{\gamma}G\alpha\tilde{\sigma}_H}{\sqrt{D^{spt}}J} + \frac{\sqrt{\gamma}\alpha l\tilde{\sigma}_G}{\sqrt{D^{spt}}J} + \frac{\sqrt{\gamma}\tilde{\sigma}_G}{\sqrt{D^{qry}J}} + \frac{\sqrt{\gamma}\alpha\tilde{\sigma}_H\tilde{\sigma}_G}{\sqrt{D^{spt}}\sqrt{D^{qry}J}}\right)$$

$$+ 8\sqrt{6}\left(\frac{\sqrt{2}\sigma_G}{\gamma T\sqrt{J}} + \frac{\sqrt{\gamma}\alpha\sigma_H G}{\sqrt{J}} + \frac{\sqrt{\gamma}\alpha l\sqrt{2}\sigma_G}{\sqrt{J}} + \frac{\sqrt{\gamma}\alpha\sqrt{2}\sigma_G\sigma_H}{\sqrt{J}}\right)$$

$$+ \frac{3\Delta}{\beta T} + \frac{8\alpha l\tilde{\sigma}_G}{\sqrt{D^{spt}}} + \frac{8\beta l}{\gamma} + 8\sqrt{10}\frac{\sqrt{\gamma}\tilde{\sigma}_G}{\sqrt{D^{qry}J}} + 8\sqrt{6}\frac{\sqrt{\gamma}\sqrt{2}\sigma_G}{\sqrt{G}}$$

$$\simeq 8\sqrt{10}\left(\frac{\tilde{\sigma}_G}{(D^{qry}J)^{\frac{3}{2}}T^{\frac{1}{2}}} + \frac{\sqrt{D^{qry}}G\tilde{\sigma}_H}{\sqrt[4]{D^{spt}}T^{\frac{1}{2}}} + \frac{\sqrt{D^{qry}}l\tilde{\sigma}_G}{\sqrt[4]{D^{spt}}T^{\frac{1}{2}}} + \frac{\tilde{\sigma}_G}{T^{\frac{1}{2}}} + \frac{\tilde{\sigma}_H\tilde{\sigma}_G}{\sqrt[4]{D^{spt}}T^{\frac{1}{2}}}\right)$$

$$+ 16\sqrt{3}\left(\frac{\sigma_G}{D^{qry}J^{\frac{3}{2}}T^{\frac{1}{2}}} + \frac{\sqrt{D^{qry}}\sqrt[4]{D^{spt}}l\sigma_G}{T^{\frac{1}{2}}} + \frac{\sqrt{D^{qry}}\sqrt[4]{D^{spt}}\sigma_G\sigma_H}{T^{\frac{1}{2}}}\right)$$

$$+ \frac{8l^2\tilde{\sigma}_G}{T^{\frac{1}{2}}} + 8\sqrt{6}\frac{\sqrt{D^{spt}}\sqrt[4]{D^{spt}}\sigma_H G}{T^{\frac{1}{2}}} + \frac{3\Delta}{D^{qry}JT^{\frac{1}{4}}} + 8\sqrt{10}\frac{\tilde{\sigma}_G}{T^{\frac{1}{4}}} + 16\sqrt{3}\frac{\sqrt{D^{qry}}\sigma_G}{T^{\frac{1}{4}}} + \frac{8l\tilde{\sigma}_G}{\sqrt[4]{D^{spt}}T^{\frac{1}{4}}} + \frac{8l}{T^{\frac{1}{4}}}$$

$$= \left(l^2\tilde{\sigma}_G + \tilde{\sigma}_G + \sqrt{D^{qry}}\sqrt[4]{D^{spt}}(\sigma_H G + l\sigma_G + \sigma_G\sigma_H)\right)\frac{1}{T^{\frac{1}{2}}}$$

$$+ \left(\sqrt{D^{qry}}G\tilde{\sigma}_H + \sqrt{D^{qry}}l\tilde{\sigma}_G + \tilde{\sigma}_H\tilde{\sigma}_G\right)\frac{1}{\sqrt[4]{D^{spt}}T^{\frac{1}{2}}}$$

$$+ \left(\tilde{\sigma}_G + \sqrt{D^{qry}}\sigma_G\right)\frac{1}{(D^{qry}J)^{\frac{3}{2}}T^{\frac{1}{2}}}$$

$$+ \left(l + \tilde{\sigma}_G\sqrt{D^{qry}}\sigma_G\right)\frac{1}{T^{\frac{1}{4}}}$$

$$+ \frac{\Delta}{D^{qry}J}\frac{1}{T^{\frac{1}{4}}} + \frac{l\tilde{\sigma}_G}{\sqrt[4]{D^{spt}}}\frac{1}{T^{\frac{1}{4}}}. \tag{60}$$

Thus we can conclude that

$$\frac{1}{T}\sum_{t=0}^{T-1}\mathbb{E}\left[||\nabla F(\boldsymbol{\theta}_t)||\right] \leq \mathcal{O}\left(\frac{\mathcal{C}_1}{T^{\frac{1}{4}}} + \frac{\mathcal{C}_2}{T^{\frac{1}{2}}}\right),$$

where $\mathcal{C}_1 = \max\{\frac{\Delta}{D^{qry}J}, l + \sqrt{D^{qry}}\tilde{\sigma}_G\sigma_G, \frac{l\tilde{\sigma}_G}{\sqrt[4]{D^{spt}}}\}$, and $\mathcal{C}_2 = \max\{(l^2 + 1)\tilde{\sigma}_G + \sqrt{D^{qry}}\sqrt[4]{D^{spt}}(\sigma_H G + l\sigma_G + \sigma_G\sigma_H), \frac{\sqrt{D^{qry}}G\tilde{\sigma}_H + \sqrt{D^{qry}}l\tilde{\sigma}_G + \tilde{\sigma}_H\tilde{\sigma}_G}{\sqrt[4]{D^{spt}}}, \frac{\tilde{\sigma}_G + \sqrt{D^{qry}}\sigma_G}{(D^{qry}J)^{3/2}}\}$.

(Inner-loop update stepsize, Outer-loop stepsize and Momentum decaying weight)

### B.4 CONVERGENCE OF FOMAML

The goal gradient is written as

$$\nabla F(\boldsymbol{\theta}) = \frac{1}{I}\sum_{i\in\mathcal{I}}\left[\left(\mathbf{I} - \alpha\nabla^2 f_i(\boldsymbol{\theta})\right)\cdot\nabla f_i\left(\theta - \alpha\nabla f_i(\boldsymbol{\theta})\right)\right]. \tag{61}$$

The FOMAML outer-loop gradient is

$$\nabla\hat{F}_1(\boldsymbol{\theta}) = \frac{1}{J}\sum_{j\in\mathcal{I}_t}\nabla f_j\left(\boldsymbol{\theta} - \alpha\nabla f_j\left(\theta;\mathcal{D}_j^{spt}\right);\mathcal{D}_j^{qry}\right). \tag{62}$$

Likely, we have $\mathbb{E}\left[\|\boldsymbol{\epsilon}_1\|^2\right]$:

$$\mathbb{E}\left[\|\boldsymbol{\epsilon}_1\|^2\right] = \mathbb{E}\left[\left\|\nabla\hat{F}_1(\boldsymbol{\theta}) - \nabla F(\boldsymbol{\theta})\right\|^2\right]. \tag{63}$$

---

**Algorithm 2:** First-Order MAML Algorithm with Normalized Momentum Update

---

**Input:** $I$ Training tasks $\mathcal{D}_i, i \in \{1, \cdots, I\}$ ; Inner loop step size $\alpha$; Outer loop step size $\beta$; Momentum parameter $\gamma$; Number of iterations $T$

**Output:** Meta-trained network initialization $\theta_T$

---

1   *Meta-training stage*
2   Initialize $\theta$ randomly.
3   **for** $t \in \{0, \cdots, T-1\}$ **do**
4      Uniformly sample $J$ meta-training tasks with index set $\mathcal{I}_t$.
5      **forall** $j \in \mathcal{I}_t$ **do**
6          Initialize the network parameter $\theta_t$.
7          Uniformly sample $D^{spt}$ data samples from $\mathcal{D}_j$ as the support set.
8          Draw another $D^{qry}$ samples from $\mathcal{D}_j$ as the query set with $D^{qry} \gg D^{spt}$.
9          Inner-loop update $\phi_j = \theta_t - \alpha\nabla f_j(\theta_t; \mathcal{D}_j^{spt})$ using the support set $\mathcal{D}_j^{spt}$.
10         Evaluate loss $f_j(\phi_j; \mathcal{D}_j^{qry})$ on query set $\mathcal{D}_j^{spt}$.
11      Compute outer loop global gradient: $\nabla\hat{F}_1(\theta_{t+1}) = \frac{1}{J}\sum_{j\in\mathcal{I}_t}\nabla f_j\left(\phi_j; \mathcal{D}_j^{qry}\right)$ .
12      Perform momentum update: $g_{t+1} = (1-\gamma)g_t + \gamma\nabla\hat{F}_1(\theta_{t+1})$ .
13      Perform normalized outer loop update: $\theta_{t+1} \leftarrow \theta_t - \frac{\beta}{\|g_{t+1}\|}g_{t+1}$.
14   *Finetuning stage*
15   Load the initialized network parameter $\theta_T$.
16   Sample $D$ data points from new tasks $\mathcal{D}_{test}$ as the support set $\mathcal{D}_{test}^{spt}$ for finetuning. The remaining data of $\mathcal{D}_{test}$ consists query set $\mathcal{D}_{test}^{qry}$.
17   Evaluate loss $f_{test}(\theta_T; \mathcal{D}_{test}^{spt})$ on support set $\mathcal{D}_{test}^{spt}$. Finetune the parameters via gradient descent as $\theta_{test} \leftarrow \theta_T - \alpha\nabla f_{test}(\theta_T; \mathcal{D}_{test}^{spt})$.

---

**Lemma 10.** *Bound on* $\mathbb{E}\left[\|\epsilon_1\|^2\right]$ *Based on the Assumptions, we have*

$$\mathbb{E}\left[\|\epsilon_1\|^2\right] \le \frac{2}{J}\left(\alpha^2 l^2 G^2 + \frac{2\alpha^2\tilde{\sigma}_G^2}{D^{spt}} + \frac{2\tilde{\sigma}_G^2}{D^{qry}}\right). \tag{64}$$

*Proof.* Proof sketch of Lemma 10 Since the proof techniques of FOMAML is similar to that of MAML, we provide only the proof sketch as following. First, following the deduction process of MAML, we also define the intermediate gradient

$$\nabla\tilde{F}_1(\theta) = \frac{1}{J}\sum_{j\in\mathcal{I}_t}\nabla f_j\left(\theta - \alpha\nabla f_j(\theta)\right), \tag{65}$$

which get rids of the data sample stochasticity. With the above-mentioned definitions can decompose (63) as follows

$$\mathbb{E}\left[\left\|\nabla\hat{F}_1(\theta) - \nabla F(\theta)\right\|^2\right] \le 2\mathbb{E}\left[\left\|\nabla\hat{F}_1(\theta) - \nabla\tilde{F}_1(\theta)\right\|^2\right] + 2\mathbb{E}\left[\left\|\nabla\tilde{F}_1(\theta) - \nabla F(\theta)\right\|^2\right]. \tag{66}$$

**Lemma 11.** *Bound on* $\mathbb{E}\left[\left\|\nabla\hat{F}_1(\theta) - \nabla\tilde{F}_1(\theta)\right\|^2\right]$.

$$\mathbb{E}\left[\left\|\nabla\hat{F}_1(\theta) - \nabla\tilde{F}_1(\theta)\right\|^2\right] \le \frac{2}{J}\left(\frac{\alpha^2\tilde{\sigma}_G^2}{D^{spt}} + \frac{\tilde{\sigma}_G^2}{D^{qry}}\right). \tag{67}$$

**Lemma 12.** *Bound on* $\mathbb{E}\left[\left\|\nabla\tilde{F}_1(\boldsymbol{\theta}) - \nabla F(\boldsymbol{\theta})\right\|^2\right]$.

$$\mathbb{E}\left[\left\|\nabla\tilde{F}_1(\boldsymbol{\theta}) - \nabla F(\boldsymbol{\theta})\right\|^2\right] \leq \frac{1}{J}\alpha^2 l^2 G^2. \tag{68}$$

Based on Lemma 12 and 11, we plug them in (66), which will finally result in Lemma 10. $\qquad\square$

Also we need no bound $\|\mathbb{E}\left[\boldsymbol{\epsilon}\right]\|$:

**Lemma 13.** ***Bound on*** $\|\mathbb{E}\left[\boldsymbol{\epsilon}\right]\|$***.*** *The norm of the expected error is bounded as*

$$\|\mathbb{E}\left[\boldsymbol{\epsilon}_1\right]\| \leq \sqrt{2}\left(\frac{\alpha l\tilde{\sigma}_G}{\sqrt{D^{spt}}} + \alpha lG\right). \tag{69}$$

Based on Lemma 10 and 13 and recall that from Lemma 5 we have

$$\frac{1}{T}\sum_{t=0}^{T-1}\mathbb{E}\left[\|\boldsymbol{e}_t\|\right] < \frac{1}{\gamma T}\mathbb{E}\left[\|\boldsymbol{e}_0\|\right] + \sqrt{\gamma}\sqrt{\mathbb{E}\left[\|\boldsymbol{\epsilon}\|^2\right]} + \|\mathbb{E}\left[\boldsymbol{\epsilon}\right]\| + \frac{1}{\gamma}\mathbb{E}\left[\|S\|\right]$$

$$\leq \left(\frac{1}{\gamma T} + \sqrt{\gamma}\right)\sqrt{\frac{2}{J}}\left(\alpha lG + \frac{\sqrt{2}\alpha\tilde{\sigma}_G}{\sqrt{D^{spt}}} + \frac{\sqrt{2}\sigma^G}{\sqrt{D^{qry}}}\right) + \sqrt{2}\left(\frac{\alpha l\tilde{\sigma}_G}{\sqrt{D^{spt}}} + \alpha lG\right) + \frac{\beta L}{\gamma}. \tag{70}$$

Finally following (22):

$$\frac{1}{T}\sum_{t=0}^{T-1}\mathbb{E}\left[\|\nabla F(\boldsymbol{\theta}_t)\|\right] \leq \frac{3\Delta}{\beta T} + \frac{8}{T}\sum_{t=0}^{T-1}\mathbb{E}\left[\|\boldsymbol{e}_t\|\right] + \frac{3\beta L}{2}$$

$$\leq \frac{3\Delta}{\beta T} + \left(\frac{8}{\gamma T} + 8\sqrt{\gamma}\right)\sqrt{\frac{2}{J}}\left(\alpha lG + \frac{\sqrt{2}\alpha\tilde{\sigma}_G}{\sqrt{D^{spt}}} + \frac{\sqrt{2}\tilde{\sigma}_G}{\sqrt{D^{qry}}}\right)$$

$$+ 8\sqrt{2}\left(\frac{\alpha l\tilde{\sigma}_G}{\sqrt{D^{spt}}} + \alpha lG\right) + \frac{8\beta L}{\gamma} + \frac{3\beta L}{2}$$

$$\leq \frac{3\Delta}{D^{qry}JT^{\frac{1}{4}}} + \frac{16\tilde{\sigma}_G}{(D^{qry}J)^{\frac{3}{2}}T^{\frac{1}{2}}} + 8\sqrt{2}\frac{\sqrt{D^{qry}}\sqrt[4]{D^{spt}}lG}{T^{\frac{1}{2}}}$$

$$+ 16\frac{\sqrt{D^{qry}}\tilde{\sigma}_G}{\sqrt[4]{D^{spt}}T^{\frac{1}{2}}} + \frac{16\tilde{\sigma}_G}{T^{\frac{1}{4}}} + 8\sqrt{2}\left(\frac{l\tilde{\sigma}_G}{\sqrt[4]{D^{spt}}T^{\frac{1}{4}}} + \frac{\sqrt[4]{D^{spt}}lG}{T^{\frac{1}{4}}}\right) + \frac{8l}{T^{\frac{1}{4}}},$$

$$\simeq \left(\tilde{\sigma}_G + l + \sqrt[4]{D^{spt}}lG\right)\frac{1}{T^{\frac{1}{4}}}$$

$$+ \frac{l\tilde{\sigma}_G}{\sqrt[4]{D^{spt}}T^{\frac{1}{4}}} + \frac{\Delta}{D^{qry}JT^{\frac{1}{4}}}$$

$$+ \frac{\sqrt{D^{qry}}\sqrt[4]{D^{spt}}lG}{T^{\frac{1}{2}}}$$

$$+ \frac{\sqrt{D^{qry}}\tilde{\sigma}_G}{\sqrt[4]{D^{spt}}T^{\frac{1}{2}}} + \frac{\tilde{\sigma}_G}{(D^{qry}J)^{\frac{3}{2}}T^{\frac{1}{2}}}. \tag{71}$$

the third inequation is obtained by setting the stepsizes as $\alpha = \frac{\sqrt[4]{D^{spt}}}{T^{\frac{1}{4}}}$, $\beta = \frac{D^{qry}\times J}{T^{\frac{3}{4}}}$ and $\gamma = \frac{D^{qry}\times J}{T^{\frac{1}{2}}}$. Likely, we conclude that

$$\frac{1}{T}\sum_{t=0}^{T-1}\mathbb{E}\left[\|\nabla F(\boldsymbol{\theta}_t)\|\right] \leq \mathcal{O}\left(\frac{\mathcal{C}_3}{T^{\frac{1}{4}}} + \frac{\mathcal{C}_4}{T^{\frac{1}{2}}}\right), \tag{72}$$

where $\mathcal{C}_3 = \max\{\frac{\Delta}{D^{qry}J}, \quad \tilde{\sigma}_G + l + \sqrt[4]{D^{spt}lG}, \quad \frac{l\tilde{\sigma}_G}{\sqrt[4]{D^{spt}}}\}$, $\mathcal{C}_4 = \max\{\sqrt{D^{qry}}\sqrt[4]{D^{spt}}lG, \frac{\sqrt{D^{qry}}\tilde{\sigma}_G}{\sqrt[4]{D^{spt}}}, \frac{\tilde{\sigma}_G}{(D^{qry}J)^{\frac{3}{2}}}\}$.

## C   SUPPLEMENTARY EXPERIMENTS

### C.1   GRADIENT NORM DECAYING TRENDS

We compare the momentum decay trends of our proposed TFMAML and TFFOMAML with the gradient decay trends of the benchmark methods, using the same experimental setup as in the main numerical results. Our findings reveal that the tuning-free scheme consistently achieves a lower gradient norm, attributable to its effective control of the bias. Although some fluctuations remain, these can be further moderated by adjusting the step sizes.

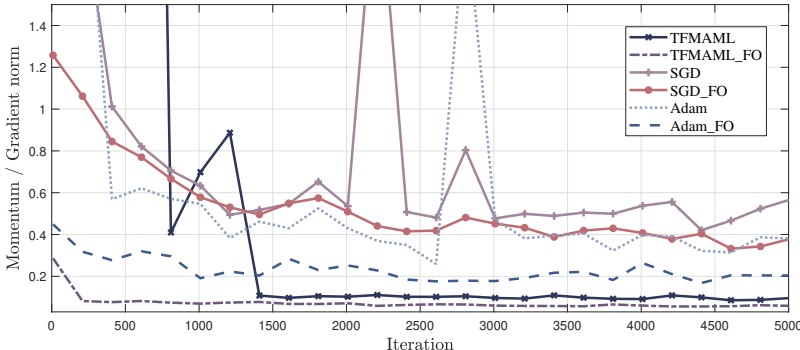

Figure 2: Norm of the momentum in MAML and FOMAML (used for the final global loss update) compared with the gradient norms of benchmark methods with 5,000 iterations.

Surprisingly, our results indicate that the first-order variants consistently exhibit more favorable gradient norm decay than their full second-order counterparts. At first glance, this seems counterintuitive: since FOMAML is only an approximation to MAML, one might expect it to converge more slowly or exhibit worse gradient behavior (Finn et al., 2017; Fallah et al., 2020a). However, the numerical result reveals several important factors:

1. **Hessian amplification of noise.** The Hessian term $(\mathbf{I} - \alpha\nabla^2 f_j)$ can magnify stochasticity in the gradient, particularly under few-shot settings where both sample size and task-batch size are small. This effect introduces larger variance in the meta-gradient and slows down the apparent decay of its norm. By skipping the Hessian, FOMAML avoids this amplification, producing smoother decay curves.

2. **Implicit regularization of first-order methods.** First-order updates, by ignoring curvature corrections, act as a biased but more stable estimator. This bias may inadvertently regularize the optimization dynamics, preventing overly aggressive updates along high-curvature directions where MAML's second-order term may destabilize training.

In summary, the results suggest that the inclusion of Hessian terms introduces a *variance–bias trade-off*: MAML is more faithful to the true gradient but more sensitive to stochasticity, whereas FOMAML sacrifices some accuracy for stability and smoother gradient decay. This aligns with both our theoretical understanding and the numerical evidence, highlighting that gradient norm dynamics and final accuracy can exhibit different behaviors.

### C.2   NUMERICAL RESULTS ON MINIIMAGENET

The MiniImagenet dataset (Ravi & Larochelle, 2017) consists of 64 training classes, 12 validation classes, and 24 test classes. Following the standard protocol in Finn et al. (2017), we use a four-layer convolutional network with 32 filters per layer to mitigate overfitting. The learning objective is the cross-entropy loss between predicted and true labels.

To ensure consistency with our theoretical analysis, we train and evaluate all methods using one inner-loop gradient step. (In contrast, the original MAML implementation Finn et al. (2017) uses multiple inner updates.) As in Finn et al. (2017), each task uses one example per class for adaptation and 15 examples per class for computing the post-update meta-gradient. We adopt a meta batch-size of 4 for 1-shot and 2 for 5-shot training. All models are trained for 60,000 iterations on a single NVIDIA RTX 3090 GPU. The resulting performance is reported in Fig. 3.

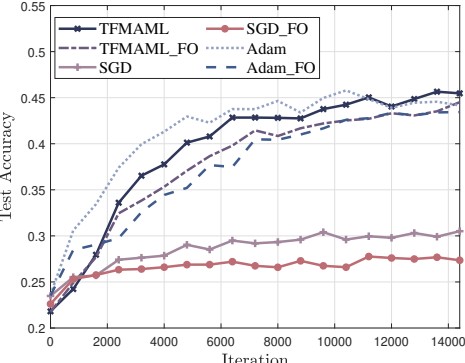

Figure 3: miniImagenet Result

## C.3 ABLATION STUDY

In this section, we evaluate the individual contributions of momentum (MM) and gradient normalization (GN) by examining both the training-time gradient norm trajectory and the resulting test accuracy. The experimental setup is identical to that used in Section 5 on the Omniglot dataset.

Figure 4 shows the evolution of the gradient norm during training. Comparing SGD with GN-only, we observe that GN alone does not significantly alter the gradient-norm behavior. In contrast, introducing momentum leads to a substantial reduction in the gradient norm, indicating its dominant role in stabilizing and accelerating the optimization dynamics.

Figure 5 reports the corresponding test accuracies. While the GN-only variant attains accuracy comparable to TFMAML, it converges noticeably slower due to the absence of momentum accumulation. Adding GN on top of momentum further improves both convergence speed and final accuracy. Overall, the combination of momentum + gradient normalization yields the fastest and most accurate performance among the tested configurations.

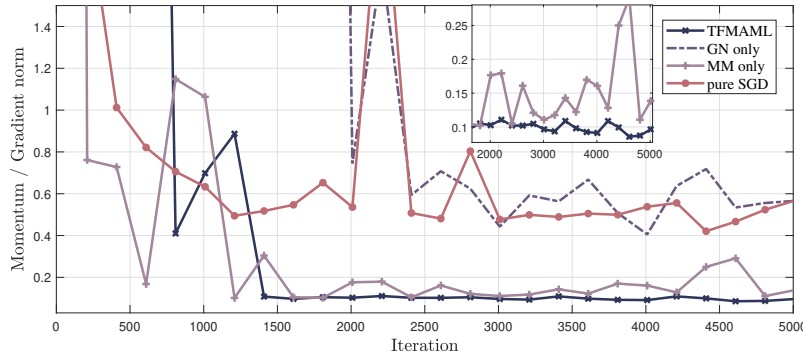

Figure 4: Ablation study on gradient norm. MM: momentum, GN: gradient normalization.

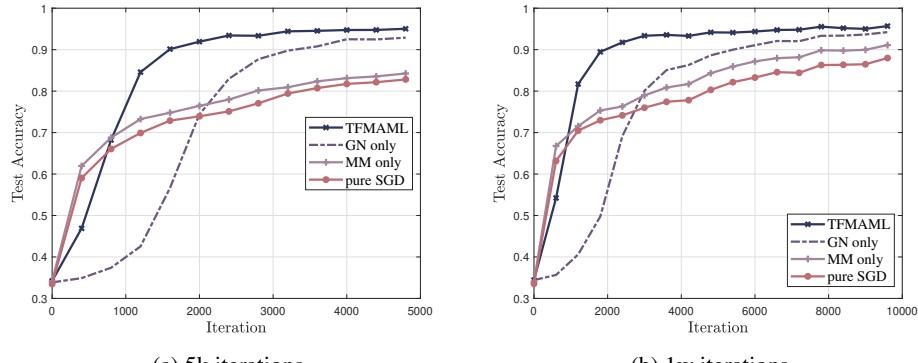

(a) 5k iterations          (b) 1w iterations

Figure 5: Ablation study versus the number of iterations on the Omniglot dataset.

## D    STATEMENT ON THE USE OF LARGE LANGUAGE MODELS (LLMs)

We disclose that the LLMs were only used to polish the writing and grammar in this paper.

