# OpenReview forum: "Problem-Parameter-Agnostic MAML"
_ICLR.cc/2026/Conference — Submitted to ICLR 2026_

### Official Review · Reviewer_PtYt · 2025-10-26

**Soundness:** 3
**Presentation:** 3
**Contribution:** 2
**Rating:** 6
**Confidence:** 3

**Summary:**

This paper considers the problem of optimization of the model-agnostic meta-learning objective, which is challenging for two reasons: 1) the nested structure and 2) the stochasticity of the inner update. Previous analyses focus on stochastic gradient descent, which rely on problem-specific learning rates and thus are difficult to apply in practice. This work proposes to use the normalized stochastic gradient descent with momentum, which enables convergence guarantees that 1) do not rely on problem-specific learning rates, step-sizes, etc. and 2) has bounds that converge to zero as the number of optimizations goes to infinity. Experimental results on Omniglot with a 4-layer CNN show improved performance over SGD and Adam after a small number of iterations.

**Strengths:**

- The paper is well-written, with the main ideas clearly emphasized.
- The proposed algorithm is simple and straightforward to apply in practice.
- The theoretical analyses appear rigorous and sound (although I did not check the math).
- The experiments validate the theoretical results and the effectiveness of the algorithm over previous works.

**Weaknesses:**

I believe that the main weakness of the paper is in its novelty.
- The proposed algorithm is a naive application of the normalized SGD with momentum framework from Cutkosky (2023).
- The same bounds to lemmas 1 and 2 are found in Fallah et al. (2020), but the paper does not make this clear. It is not clear what the main theoretical innovations are in this paper, so I would appreciate if that is clarified.

**Questions:**

- How does the dependency on T in theorems 1 and 2 compared to SGD?
- FO Adam appears quite good compared to Adam, which is not true of TFMAML and SGD. Is there any intuition why?

---

> ### Author Response · Authors · 2025-11-19
> **Point-to-Point Responses to Reviewer PtYt**
>
> We greatly appreciate your expertise on optimization and prior works, including Cutkosky’s framework and Fallah et al. Your insights on both the theoretical framework and observed numerical phenomena prompted us to:
> 1. __Clarify our methodology and strengthen the presentation__, enhancing technical rigor and overall clarity.
> 2. Provide a more __detailed analysis of FO-Adam’s observed superiority over full Adam.
> Our point-to-point responses are as follows:
>
> ### __Reply to Weakness 1: Theoretical gap of Cutkosky's framework__
> We appreciate the reviewer’s pointer to Cutkosky’s lecture note (https://optmlclass.github.io/notes/notes15_norm.pdf), which we are familiar with. However, we respectfully disagree that TFMAML is merely a naïve application. A closer examination reveals __two fundamental obstacles__ that prevent a direct application of the normalized SGD with momentum framework to MAML. These obstacles are precisely what our work resolves:
> - __Biased stochastic gradients in MAML (violating Cutkosky’s assumptions).__ Cutkosky’s analysis relies on unbiased stochastic gradients (e.g., $\mathbb{E}[\langle r_t, r'_t\rangle]=0$, p. 3 in the lecture note). However, SGD-based MAML inherently produces biased gradients due to the inner-loop adaptation, as formally shown in __Lemma 2__ of our paper. This bias directly breaks the orthogonality condition required for Cutkosky’s momentum analysis, so his framework cannot be applied as-is.
> - __Unknown smoothness and incompatible stepsize rules__. Cutkosky’s stepsize requires access to a known smoothness constant (e.g., $\eta = \frac{\sqrt{\Delta \alpha}}{\sqrt{HT}}$, Theorem 3, p. 4). In practice, this constant is unknown for deep networks, and in MAML the global smoothness becomes even more complex because of the nested inner–outer updates, which is captured precisely in __Lemma 1__. Therefore, the prescribed stepsize cannot be used in MAML, and a direct transplant of Cutkosky’s algorithm would not guarantee convergence.
>
> In short, Cutkosky’s method cannot be naively applied to MAML because its foundational assumptions fail in the meta-learning setting. Our contribution lies exactly in resolving these issues: we account for the MAML-specific gradient bias, characterize its composite smoothness, and design inner/outer stepsizes and momentum to obtain a __problem-parameter-free, tuning-free, and provably convergent meta-optimizer__, whose capability is not provided by the original normalized SGD with momentum framework.
>
> ### __Relpy to Weakness 2: Reasons of referring to Lemma 1, 2__
> We thank the reviewer for raising this point. We have further emphasized the reason for referring to these lemmas in our revised manuscript.
>
> While it is true that Fallah et al. (2020) identify the presence of gradient bias and the dependence on a global smoothness parameter, our contribution goes substantially __beyond restating__ these observations.
>
> - In our main text, __Lemma 1 formalizes the specific structure and magnitude of the biased stochastic gradient__ in SGD-based MAML, and __Lemma 2 characterizes the compound smoothness and its the impractical dependence on unknown smoothness constants when tuning stepsizes__. These lemmas are not introduced as novel mathematical facts; rather, they serve to highlight two well-recognized but crucial obstacles that fundamentally hinder the design of theoretically grounded MAML optimizers.
>
> - Crucially, __Fallah et al. (2020) do not provide a mechanism to overcome these obstacles without relying on problem-specific parameters__, whereas the central contribution of our work is to develop an __easy-to-implement, problem-parameter-free meta-optimizer__ that _provably_ handles both challenges simultaneously. Our analysis shows how to appropriately combine normalization, momentum, and carefully chosen inner- and outer-loop stepsizes to obtain convergence guarantees __without requiring knowledge of smoothness constants, and despite the inherent gradient bias__. This yields a fully tuning-free algorithm with convergence guarantees that are not available in existing MAML or meta-optimization frameworks.
>
> In summary, with Lemmas 1 and 2 articulate known difficulties, the __core theoretical innovation__ of our work is providing the first convergence-guaranteed, problem-parameter-free solution that explicitly resolves these difficulties in the context of MAML.
>
> Cont.

---

> ### Author Response · Authors · 2025-11-19
> **Point-to-Point Responses to Reviewer PtYt**
>
> ### __Reply to Question 1: Stepsize setting and convergence of SGD-based MAML__
>
> In prior analyses of MAML, such as (Fallah, et al. 2020, Ji et al., 2022), the outer-loop stepsize is typically parameterized as:
> $ \hat{L}(\boldsymbol{\theta}) := l(1 + \alpha l)^2 + \frac{\rho \alpha (1 + \alpha l)}{J} \sum_{j \in \mathcal{I}_t} \left\| \nabla f_j (\boldsymbol{\theta}; \mathcal{D}_j^{spt}) \right\|$.
> This formulation leads to a stochastic stepsize rule $ \hat{\beta}(\boldsymbol{\theta}) = \frac{c}{\hat{L}(\boldsymbol{\theta})},
> $
> By further setting $\alpha \in (0, \frac{1}{6l}]$, SGD-based MAML achieves a convergence guarantee where:
> $\mathbb{E}\left[ \left\| \nabla F(\boldsymbol{\theta}) \right\| \right] \leq \mathcal{C} + \mathcal{O}(\frac{l}{T^{1/2}})$, where $\mathcal{C}$ in a non-diminishing constant.
>
> Key observations regarding this approach include:
> - __Stepsize Dependency__: The method relies on problem-specific smoothness parameters, limiting its generalizability.
> - __Convergence Characteristics__: A non-diminishing constant error term prevents achieving a true $\epsilon$-FOSP convergence.
>
> As detailed in Remark 4 of our manuscript, these limitations underscore the need for more adaptive and convergence-guaranteed MAML optimization strategies.
>
> ### __Reply to Question 2: Possible Explanation for FO-Adam Performing Better Than Full Adam in MAML__
> We thank the reviewer for this insightful question. This behavior has also been reported in the original MAML work (Finn et al., 2017, Sec. 5.2), where FO-MAML performs comparably—or even better—than the full second-order version. The key reason is that __Adam’s adaptive moment updates introduce additional curvature-dependent terms__ , which make the meta-gradient extremely sensitive to Hessian variation during inner-loop adaptation.
>
> * Specifically, Adam computes first and second-order momentum
> $m\_t = \beta_1 m\_{t-1} + (1-\beta\_1) g\_t,\quad
> v\_t = \beta_2 v\_{t-1} + (1-\beta_2) g\_t^2,$
> and update the network parameter as $\theta_{t + 1} = \theta\_t - \alpha \frac{m\_t}{\sqrt{v\_t}}$.
> Differentiating through the recursions requires backpropagating through both $m\_t$ and $v\_t$. Since $g\_t$ changes rapidly due to task-dependent curvature, the implicit $\frac{\partial \theta\_{t+1}}{\partial \theta\_t}
> = I - \alpha H \cdot
> \frac{\partial (m\_t / \sqrt{v\_t})}{\partial g\_t}$ become **noisy and poorly conditioned**, destabilizing the full second-order Adam meta-update. FO-Adam **removes these unstable second-order terms** while preserving Adam’s strong task-level adaptation, so it can outperform full-order Adam.
>
> __References:__
>
> __(Fallah et al., 2020)__ Fallah, A., Mokhtari, A., & Ozdaglar, A. (2020, June). On the convergence theory of gradient-based model-agnostic meta-learning algorithms. In International Conference on Artificial Intelligence and Statistics (pp. 1082-1092). PMLR.

---

> ### Comment · Reviewer_PtYt · 2025-11-28
>
> Thank you to the authors for your reply. After reading it and the other reviews, I have decided to keep the same score.

---

> > ### Author Response · Authors · 2025-12-02
> > **Thank you for your feedback**
> >
> > Thank you for taking the time to review our rebuttal and for your continued consideration. We appreciate your efforts and respect your evaluation.

---

### Official Review · Reviewer_ejKg · 2025-10-31

**Soundness:** 2
**Presentation:** 1
**Contribution:** 1
**Rating:** 2
**Confidence:** 4

**Summary:**

In this paper, the author proposes a new optimizer for meta-learning that incorporates normalized gradient updates. The work identifies that, during meta-learning updates, momentum can become unstable and introduce systematic bias. Additionally, variability in gradients makes selecting an appropriate step size challenging. To address these issues, the author introduces a framework called TFMAML, which integrates momentum aggregation with gradient normalization to better control the step size. The paper further provides theoretical analysis and empirical results demonstrating the effectiveness of the proposed method.

**Strengths:**

1: The author identifies a key challenge in optimizing meta-learning: the large variance of gradients, which makes stable updates difficult.

2: The author provides theoretical analysis that helps address the update challenge and demonstrates the importance of the proposed approach.

**Weaknesses:**

1: The paper’s presentation is difficult to follow. It frequently switches between prior work and the proposed method, making the narrative unclear. A clearer structure separating background, method, and contributions is needed.

2: The main contribution is not clearly stated. It appears the paper proposes an outer-loop optimizer for MAML using gradient normalization for step-size control, while momentum aggregation is already common in optimizers like Adam. The novelty and distinction from existing methods should be clarified.

3: Experiments rely mainly on Omniglot. Standard benchmarks such as miniImageNet and tieredImageNet should be included. Moreover, the reported Omniglot accuracy (70%–80%) is far below commonly reported MAML results (98.7%), which requires explanation.

**Questions:**

1: Could you more clearly describe your method? Specifically, where are momentum and gradient normalization applied — in the inner loop or the outer loop?

2: Typically, the outer loop of MAML uses Adam, which already incorporates momentum and adaptive step sizes. How does your approach differ from Adam in this context?

3: Could you clarify how your method addresses the limitations discussed in the paper, especially in comparison to Adam?

---

> ### Author Response · Authors · 2025-11-19
> **Point-to-Point Responses to Reviewer ejKg**
>
> We thank the reviewer for the helpful comments. Your feedback has helped us strengthen both the clarity and completeness of the paper. We have made the following improvement:
> 1. __Improved the paper’s organization__, clarifying the role of prior lemmas and better separating background from our contributions.
> 2. __Added new experiments on miniImageNet__, demonstrating the robustness and generality of TFMAML under a more challenging dataset and architecture. (See Figure 3 in the updated manuscript, or view it online at https://imgur.com/a/VqW2f62)
>
> ### __Response to Weakness 1: Organization of the paper.__
>
> Thanks for pointing this out. __We have improved our organization and emphasize on the intention of some referred lemmas in the main body of context.__ However, we would like to emphasize that, the manuscript intentionally interleaves background and method because the core theoretical obstacles, formalized in Lemma 1 and Lemma 2, are inseparable from the design of our MAML-optimizer. These two challenges, originally identified in Fallah et al. (2020), are:
> - __Biased meta-gradients__ caused by stochastic inner updates;
> - __Unknown, composite smoothness__ arising from the coupling of inner and outer loops.
> These properties make classical convergence analyses inapplicable to MAML. While Fallah et al. (2020) characterizes the difficulties, it does __not__ resolve them or provide a problem-parameter-free solution.
>
> Our work __directly overcomes both issues__ and provides the first __problem-parameter-free, provably convergent__ meta-optimizer for this setting. To improve readability, the revised version now more clearly separates background, theoretical challenges, and contributions—but the narrative remains centered around these fundamental obstacles, as this structure is essential to understanding the novelty of our approach.
>
>
> ### __Response to Weakness 2 and Question 2: TFMAML's critical difference compared with Adam__
>
> To the best of our knowledge, __TFMAML is the first work to resolve MAML’s dependence on numerous problem-specific stepsize parameters__—which are generally unknown in practice—while __guaranteeing convergence__. No existing work provides a convergence-guaranteed, tuning-free approach for MAML with __Adam__, whose hyperparameters are typically chosen via extensive trial and error (Finn et al., 2017).
>
> Specifically, __TFMAML is a convergence-guaranteed optimizer__, where inner- and outer-loop stepsizes can be computed __exactly__ based only on the number of iterations, tasks, and support/query samples. This makes TFMAML fundamentally different from prior approaches:
>
> - __Trial-and-error stepsize tuning (e.g., Adam)__: In vanilla MAML, Adam is used as a black-box optimizer with stepsizes set via empirical tuning. While this works reasonably well, finding suitable inner- and outer-loop stepsizes requires extensive grid search, and improper choices can prevent convergence altogether.
> - __Problem-parameter-based tuning__: Existing theoretical analyses, such as Fallah et al. (2020), require knowledge of smoothness parameters of the gradient and Hessian, as well as instantaneous gradient norms, which are generally __unavailable in practice__. Moreover, their SGD-based MAML analysis shows a __non-vanishing convergence gap__ with iterations. TFMAML, in contrast, __does not rely on any problem-specific parameters__ and provides guaranteed convergence.
>
> In short, the main contribution of our work is to provide a __provably convergent, parameter-free meta-optimizer for MAML__, distinguishing it from prior methods that either rely on trial-and-error tuning or require inaccessible problem-specific constants.
>
> ### __Response to Weakness 3: Extended numerical results.__
> Thank you for your suggestion. We have conducted __additional experiments on MiniImageNet__, with results presented in Appendix C.2., Figure 3.
>
> As for the reported numerical result:
> - We would like to clarify that __the reported 70–80% accuracy is not for vanilla MAML__. The 70–80% figure corresponds to __first-order SGD with only one inner-loop gradient step and 5,000 training iterations__. In contrast, the __original MAML__ (Finn et al. 2017) uses Adam as the outer-loop optimizer and __multiple inner updates__ (10 steps during training, 3 during finetuning), achieving over 95% accuracy.
> - To be consistent with our theoretical analysis, we also evaluated MAML with Adam using __only one inner update__, which similarly yields accuracy above 95%, aligning with standard results.
>
> Cont.

---

> ### Author Response · Authors · 2025-11-19
> **Point-to-Point Responses to Reviewer ejKg**
>
> ### __Response to Question 1: Clearer description of the method.__
> Both __momentum aggregation__ and __gradient normalization__ are applied __only in the outer loop__. They are not used in the inner loop because, in the nested-loop setting, applying normalization or momentum at the inner updates would involve higher-order gradients, which are prone to oscillation and difficult to compute efficiently.
>
> Following your suggestion, we have __highlighted the distinction between TFMAML and vanilla MAML__ in __Algorithm 1__ to clarify the placement of these components.
>
> ### __Response to Question 2:__
> The response to Question 2 has been incorporated into our reply for Weakness 2; please refer to that section.
>
> ### __Response to Question 3: The way we address the limitations.__
> 1. __Challenge 1 (Biased Gradient).__
>
> __Momentum aggregation__ directly suppresses the gradient bias through its recursion. As shown in Lemma 4 (Appx. B.2), the error term
> $$
> \boldsymbol{e} _t = (1 - \gamma)^t \boldsymbol{e} _0 + \sum _{\tau = 0}^{t-1} \gamma  (1 - \gamma)^{t-\tau-1} \boldsymbol{\epsilon} _{\tau+1} + \sum _{\tau = 0}^{t-1} (1-\gamma)^{t-\tau} S _{\tau},$$
>  decays geometrically, meaning the bias terms diminish over iterations. Without momentum. Without momentum, $\boldsymbol{e} _t = \boldsymbol{\epsilon} _t$ (Using the formula for the sum of a geometric series), and the bias cannot be effectively reduced. Thus, momentum is essential for addressing Challenge 1.
>
> 2. __Challenge 2 (Unknown Smoothness Constants)__
>
> Our __outer-loop gradient normalization__ eliminates the need for problem-specific smoothness parameters required in prior analyses (e.g., Fallah et al., 2020). The normalization mechanism automatically adjusts the effective step size during training, yielding a fully __problem-parameter–free__ update rule (Appendix B.2).
>
> 3. __Compared with Adam__
>
> Adam incorporates first- and second-order moments, making its behavior in the nested MAML structure analytically intractable. __No__ existing work provides convergence guarantees for Adam-based MAML, and its step sizes rely on __extensive trial-and-error__.
> In contrast, our method is the first to __provably__ resolve both the gradient bias and the step-size dependence, achieving a __tuning-free, convergence-guaranteed__ meta-optimizer that Adam does not provide.
>
> __References:__
>
> __(Fallah et al., 2020)__ Fallah, A., Mokhtari, A., & Ozdaglar, A. (2020, June). On the convergence theory of gradient-based model-agnostic meta-learning algorithms. In International Conference on Artificial Intelligence and Statistics (pp. 1082-1092). PMLR.
>
> __(Finn et al. 2017)__ Finn, C., Abbeel, P., & Levine, S. (2017, July). Model-agnostic meta-learning for fast adaptation of deep networks. In International conference on machine learning (pp. 1126-1135). PMLR.

---

### Official Review · Reviewer_nLZX · 2025-11-01

**Soundness:** 2
**Presentation:** 3
**Contribution:** 4
**Rating:** 6
**Confidence:** 3

**Summary:**

This paper proposes a method that removes the need to tune the learning rate of MAML in a multitask learning setting. MAML and its subsequent variant FOMAML are well known as representative methods in meta learning, yet theory has shown that the inner-loop step size must be tuned to problem-specific factors such as task heterogeneity and loss smoothness. The paper shows that, under certain assumptions, convergence of the proposed method can be guaranteed by scheduling the step size using quantities observable from the dataset and the problem setup.

**Strengths:**

- The logical flow is clear and well written. In particular, the problem setup and assumptions are stated explicitly, which makes the argument easy to follow.
- It is meaningful that the paper show the global-gradient estimator becomes biased due to the estimation error of the inner-loop gradient.
- The theoretical results appear sound.
- Prior work on step-size scheduling from a theoretical viewpoint required approximations using quantities such as gradient norms, whereas the proposed method uses only quantities that are evident from the problem setup. This is a solid contribution from a practical perspective.
- On Omniglot, the numerical experiments show that the proposed method achieves higher generalization performance with fewer iterations than MAML or FOMAML with naive SGD. It also achieves performance comparable to MAML or FOMAML that use Adam, which is common in practice.

**Weaknesses:**

- **Insufficient empirical validation:** While the theoretical contribution is valuable, there are concerns about whether the theoretical claims truly align with empirical facts, as follows.
    - The theoretical claims mainly discuss convergence guarantees of the proposed method, which is distinct from generalization performance. In contrast, the experiments compare the proposed method with baselines in terms of generalization, so the theory was not directly verified.
    - Lemma 2 shows that the original MAML has a bias in its global-gradient estimate. How problematic is this bias in practice was not shown. Although the paper provides an upper bound on the norm of the bias, that bound suggests the bias could be negligible in realistic regimes depending on the loss, gradient smoothness, and the choice of step size. A numerical study that measures the magnitude of gradient estimation bias under realistic settings would strengthen the paper. A similar study is also needed to show how momentum alleviates the bias.
    - From a practical viewpoint, the main theoretical contribution appears to be an algorithm that does not require hyperparameter tuning. However, the experiments essentially compare the proposed algorithm and baselines essentially on a single problem setup. To substantiate the claim that the method is truly problem-parameter-agnostic, the paper should show robust effectiveness across multiple datasets and/or model architectures.
- **Concerns about consistency in the narrative:** The introduction highlights two main issues of MAML:
    1. the presence of bias in the estimation of the global gradient, and
    2. the fact that the appropriate step size depends on problem-specific factors.

    The paper provides a clear discussion for the second point and explains why the proposed approach can resolve it. For the first point, however, it does not sufficiently explain how introducing momentum removes the estimation bias, especially the kind described in Lemma 2. The paper cites prior work to claim that momentum is helpful, but it should give a more concrete logical explanation of how momentum addresses the bias.

**Questions:**

- Reading the problem setup in Section 2, it appears that both training and test tasks are sampled from the same finite set of tasks. In the meta learning scenario targeted by MAML, one typically aims to infer unseen tasks at test time that were not provided during training. Am I correct that the problem setting in this paper is multitask learning rather than meta learning? Since the abstract and other parts describe the setting as meta learning, I found this somewhat misleading.
- Related to the previous question, meta learning is essentially about generalization to unseen tasks. From that viewpoint, how do guarantees on the time-averaged gradient norm converging over iterations contribute to generalization performance?
- In line 112, since $\mathcal{I}$ is a set, should it not be $\mathcal{I} = \\{1, \ldots, I\\}$? It seems the curly braces are missing.
- As noted in the experimental section, practitioners typically use Adam rather than plain SGD. Can any discussion or extensions be derived from the theoretical results to cover such optimizers?

---

> ### Author Response · Authors · 2025-11-19
> **Point-to-Point Responses to Reviewer nLZX**
>
> We thank the reviewer for the detailed feedback, especially regarding the role of momentum in reducing the bias—an important aspect both theoretically and empirically. Based on these insightful comments, we have improved the paper in the following ways:
> 1. __Added experiments__ showing how momentum accelerates gradient-norm decay and improves convergence in accuracy. (See Figures 4 and 5 in the updated manuscript, or view them online at https://imgur.com/a/sOzzZY0)
> 2. __Included results on MiniImageNet__ with a different model architecture to demonstrate the method’s generalizability.
> (See Figure 3 in the updated manuscript, or view it online at https://imgur.com/a/VqW2f62)
> 3. __Clarified the mechanism__ by which momentum mitigates the effect of biased meta-gradient estimates.
>
> Our point-to-point responses are as follows:
>
> ### __Response to Weakness 1: Strengthened Empirical Evidence__
>
> - __1.1 Gradient norm converging curve__: We acknowledge that there is a gap in the convergence guarantees and the generalization performance in terms of accuracy. This is a common gap in many of the existing works, since the accuracy on the test set is the main concern in practice and can indirectly reveal the performance of the trained model. __To more closely align theory and experiments, we additionally report the empirical gradient-norm convergence curves (Fig. 2), which verify the predicted optimization behavior of our method.__ This directly demonstrates that the proposed meta-optimizer behaves as guaranteed by our theory while achieving superior generalization in practice.
>
> - __1.2 Illustration of how momentum alleviates the bias__: We appreciate the reviewer’s suggestion, but quantitatively measuring the gradient-estimation bias in MAML is not feasible: computing the ''unbiased'' meta-gradient requires evaluating the exact outer gradient over all tasks and all samples, which is intractable for few-shot benchmarks. Consequently, the bias term itself cannot be computed or visualized directly in realistic settings.
> Instead, we evaluate the observable effect of bias on the optimization dynamics. __Fig. 4__ shows that adding momentum substantially reduces the gradient norm and stabilizes training, while __Fig. 5__ demonstrates faster and higher-accuracy convergence. These behaviors are exactly what our theory predicts when momentum counteracts biased gradient directions. Together, these results provide clear empirical evidence that momentum meaningfully mitigates the practical impact of MAML’s gradient bias.
>
> - __1.3 Experiment on other datasets and model architectures__: We additionally conducted experiments on the more challenging __MiniImageNet__ dataset, which has significantly higher-dimensional inputs and a different network architecture than Omniglot. As shown in __Fig. 3__, the proposed TFMAML consistently outperforms all baselines under this more complex setting as well, demonstrating that our method generalizes robustly across datasets and architectures.
>
>
> ### __Response to Weakness 2: Explaining the effect of momentum in reducing the bias__
>
> TFMAML updates outher-loop parameters via:
> $$
> \boldsymbol{g} _{t+1} = (1-\gamma)\boldsymbol{g} _{t} + \gamma \nabla \hat{F}(\boldsymbol{\theta} _{t+1}), \quad \boldsymbol{\theta} _{t + 1} = \boldsymbol{\theta} _t - \frac{\beta}{||\boldsymbol{g} _t||} \cdot \boldsymbol{g} _t,\\
> $$
>
> From the normalized update, we obtain (Lemma 3, Supplementary Material):
> $$
> \frac{1}{T}\sum _{t = 0}^{T-1} \mathbb{E}\left[|| \nabla F(\boldsymbol{\theta} _t) || \right] \leq \frac{3\Delta}{\beta T} + \frac{8}{T}\sum _{t = 0}^{T-1} \mathbb{E}\left[|| \boldsymbol{e} _t ||\right] + \frac{3\beta L}{2} ,
> $$
>
> where $\boldsymbol{e} _t := \boldsymbol{g} _t - \nabla F(\boldsymbol{\theta} _t)$. Thus, controlling the bias term $\mathbb{E}\left[|| \boldsymbol{e} _t ||\right]$ is essential.
>
> Define the instantaneous stochastic-gradient error $\boldsymbol{\epsilon} _t = \nabla \hat{F}(\boldsymbol{\theta} _t) - \nabla F(\boldsymbol{\theta} _t)$. __Using the momentum recursion__ (Lemma 4), we obtain:
> $$
> \boldsymbol{e} _t = (1 - \gamma)^t \boldsymbol{e} _0 + \sum _{\tau = 0}^{t-1} \gamma  (1 - \gamma)^{t-\tau-1} \boldsymbol{\epsilon} _{\tau+1} + \sum _{\tau = 0}^{t-1} (1-\gamma)^{t-\tau} S _{\tau},
>  $$
> This decomposition shows that __momentum exponentially damps both the accumulated stochastic-gradient noise and the structural MAML bias__, since each term is multiplied by $(1-\gamma)$ with a small $\gamma$. In contrast, __without momentum__, $\boldsymbol{e} _t = \boldsymbol{\epsilon} _t$, and the bias cannot be reduced.
>
> Cont.

---

> ### Author Response · Authors · 2025-11-19
> **Point-to-Point Responses to Reviewer nLZX**
>
> ### __Response to Question 1: Seperated training and test dataset__
> Thank you for pointing this out. We would like to clarify that Section 2 describes only the __training process__, where support (fine-tuning) and query (meta-update) samples are drawn from the same training task, exactly following the original MAML setups (Finn et al., 2017). However, in our experiments, __training and testing tasks are strictly separated__. For Omniglot, 1,200 characters are used for training and the remaining 423 for testing, with __no overlap__. All evaluation is performed on characters never seen during training.
> Following your suggestion, we have __explicitly clarified this separation in the introduction__.
>
> ### __Response to Question 2: Convergence guarantees and generalization ability__
> We respectfully clarify that convergence in the time-averaged expected gradient norm is __a standard and appropriate criterion for non-convex meta-optimization__; for example, it is precisely the metric used to establish convergence in STORM (Cutkosky & Orabona, 2019) and FedCM (Xu et al., 2021). Our result therefore guarantees that the algorithm approaches an approximate first-order stationary point of the empirical meta-objective, which is a strong tractable notion available in __general non-convex settings__.
> Moreover, the connection between such __optimization guarantees and meta-generalization is well supported by stability analyses for non-convex stochastic optimization__ (Kuzborskij & Lampert, 2018; Lei & Ying, 2020) and by meta-learning generalization studies (Amit & Meir, 2018). These works show that controlling the expected gradient norm, together with standard smoothness and boundedness assumptions, leads to a provably small gap between empirical and population meta-risk.
>
> ### __Response to Question 3: Notation of set__
> Thank you for pointing this out, we have corrected this definition in the paper.
>
> ### __Response to Question 4: Discussion on Adam__
> We agree that Adam is widely used in practice; but __extending our theory to Adam is fundamentally nontrivial__: most existing Adam analyses (even in single-task, nonconvex settings) require strong assumptions or problem-dependent parameters. A tuning-free, problem-parameter-free convergence guarantee for Adam-based MAML algorithm is still largely open.
>
> __References:__
>
> __(Finn. et al, 2017)__ Finn, C., Abbeel, P., & Levine, S. (2017, July). Model-agnostic meta-learning for fast adaptation of deep networks. In International conference on machine learning (pp. 1126-1135). PMLR.
>
> __(Cutkosky & Orabona, 2019)__ Cutkosky, A., & Orabona, F. (2019). Momentum-based variance reduction in non-convex sgd. Advances in neural information processing systems, 32.
>
> __(Xu et al., 2021)__ Xu, J., Wang, S., Wang, L., & Yao, A. C. C. (2021). Fedcm: Federated learning with client-level momentum. arXiv preprint arXiv:2106.10874.
>
> __(Kuzborskij & Lampert, 2018)__ Kuzborskij, I., & Lampert, C. (2018, July). Data-dependent stability of stochastic gradient descent. In international conference on machine learning (pp. 2815-2824). PMLR.
>
> __(Lei & Ying, 2020)__ Lei, Y., & Ying, Y. (2020, November). Fine-grained analysis of stability and generalization for stochastic gradient descent. In International Conference on Machine Learning (pp. 5809-5819). PMLR.
>
> __(Amit & Meir, 2018)__ Amit, R., & Meir, R. (2018, July). Meta-learning by adjusting priors based on extended PAC-Bayes theory. In International Conference on Machine Learning (pp. 205-214). PMLR.

---

> > ### Comment · Reviewer_nLZX · 2025-11-27
> >
> > Thank you very much for your detailed answer.
> >
> > > Response to Weaknesses 1/2
> >
> > Thank you very much for your clarification. The explanation and additional experiments addressed my concerns.
> >
> > > Response to Question 1: Seperated training and test dataset
> >
> > It seems my explanation wasn't clear enough, and the intended meaning didn't get across properly. I apologize for that. Let me clarify my concerns again. What I'm concerned about isn't whether numerical experiments are being conducted under task holdout conditions, but rather what kind of problem formulation is being provided for theoretical guarantees.
> >
> > Based on the content from line 112, I understand this paper is considering a distribution over **a finite set of tasks** $I = \\{ 1, \ldots, I \\}$. Further, as stated in line 125 ("Let $\mathcal{I}\_t \subset \mathcal{I}$ denote the task index set sampled at iteration $t$"), the problem setup appears to involve randomly subsampling subsets of the entire task set through some method at each iteration. Primarily, this suggests the theoretical analysis is being conducted within a multi-task learning framework where **the model is expected to observe all tasks after sufficient training time**. If my understanding contains any errors, please point out the mistake. While the algorithm under analysis is indeed a commonly employed meta-learning algorithm and the experimental setup follows this convention, the theoretical analysis seems distinct from typical meta-learning settings that consider continuous task spaces, for example.
> >
> > ---
> >
> > > Response to Question 2: Convergence guarantees and generalization ability
> >
> > This was not intended to imply any concerns about the convergence guarantee itself. Building on the above understanding, I'm seeking discussion about how the convergence guarantees provided in a multi-task learning ($\neq$ meta-learning) setting might contribute to generalization to unseen tasks in the meta-learning sense.
> >
> > Regarding the connection to meta-learning, the Amit & Meir 2018 paper you cited considers a setting where the task distribution $\mathcal{D}$ itself is generated iid from an unknown task distribution $\tau$ (as shown in Equation 1). This appears to create a conceptual gap with your current paper, which only assumes a finite base set of tasks.
> >
> > ---
> >
> > > Response to Question 4: Discussion on Adam
> >
> > Thank you very much for your clarification. I fully understand that while theoretical analysis typically employs SGD, numerical experiments often use Adam, and that generalizing to Adam is not straightforward. I do not believe this point alone would determine whether a paper is accepted or rejected. Nevertheless, I commented because I thought that if any discussion on this matter were possible, it might offer practical insights.

---

> > > ### Author Response · Authors · 2025-12-01
> > > **Follow-up Response to Reviewer nLZX**
> > >
> > > ### __Follow-up Response 1: Clarifying Our Meta-Learning Framework vs. Multi-Task Learning (MLT)__
> > >
> > > Thank you for follow-up questions. We first clarify that __our work is strictly based on the MAML framework__, not MLT. The core objective in MAML, also in our algorithm, theory, and experiments, is to __learn an initialization that enables rapid adaptation to previously unseen tasks using only a few data samples and a few gradient steps__. This is fundamentally different from MTL, whose goal is to jointly learn multiple fixed tasks observed during training.
> > >
> > > - __Our training and analysis setup follows standard few-shot MAML, not MTL.__ Using Omniglot as an example:
> > >     - The meta-learner must quickly adapt to __new, unseen 5-way classification tasks__. The specific semantic labels (cat, dog, apple, etc.) are irrelevant; what matters is distinguishing new classes using virtual labels $\{1, \cdots, 5\}$. Even with 1200 training characters, the number of possible 5-way tasks is $C_{1200}^5 \approx 2.4 \times 10^{15}$, yet typical MAML training enumerates only about $4 \times 10^{5}$ tasks.
> > >     - __Therefore, the model cannot and is not expected to see all possible tasks during training.__ This matches the MAML objective: __generalization to unseen tasks drawn from the same task distribution__, not convergence on a fixed finite set of tasks as in MTL. Consequently, our framework is consistent with the standard MAML in Finn et al. (2017) and similar to the setup in Amit & Meir (2018), where
> > >         >  ...the task distribution $\mathcal{D}$ itself is generated iid from an unknown task distribution $\tau$...
> > >     - Why this is not MTL: MTL assumes a fixed, finite set of tasks, and the model eventually observes all of them. __This assumption does not hold in MAML__, where the space of possible tasks is combinatorially large and fundamentally unobservable in full.
> > >
> > > - __Second, the fundamental algorithmic distinction: MTL vs. MAML gradient updates.__ A core difference between MTL and MAML lies in their optimization objectives and update rules:
> > >     1. __MTL__ trains a single shared model that must perform well on __all observed training tasks simultaneously__, focusing on shared representations rather than rapid adaptation. The objective is (under stochastic training with minibatches of tasks) $\sum_{i=1}^I f_i (\theta, \mathcal{D}^{batch}_i).$
> > >     The model eventually sees __all__ tasks in the training set, Its success relies on adequate coverage of these tasks, consistent with the MTL assumption of a fixed, finite task set.
> > >
> > >     2. __MAML__ instead optimizes for __fast adaptation__. Each task induces a task-specific inner-loop update $\theta_j =  \theta - \alpha \nabla f_i(\theta, \mathcal{D}_i^{spt})$, and the meta-objective becomes $\sum\_{i}^I f\_i(\theta - \alpha \nabla f\_i (\theta, \mathcal{D}^{spt}\_i), \mathcal{D}^{qry}\_i)$ (Not required to enumerate all the tasks).
> > >     Thus, MAML optimizes not for joint performance on observed tasks, but for an initialization $\theta$ that enables __rapid adaptation to new, unseen tasks__ after just 1–5 gradient steps. In stochastic training, each iteration samples only a _tiny fraction_ of all possible tasks, each with few support samples, aligning precisely with the MAML setting where tasks are drawn i.i.d. from a large underlying task distribution.
> > >
> > > ### __Follow-up Response 2: Extended Discussion on Adam__
> > >
> > > Thank you for raising this point. Some of our high-level analysis is as follow:
> > > - __Why Adam is difficult to analyze in a problem-agnostic way.__
> > > Adam maintains first- and second-order moments
> > > $m_t = \beta_1 m_{t-1} + (1-\beta_1) g_t,\quad v_t = \beta_2 v_{t-1} + (1-\beta_2) g_t^2,$
> > > and update as $\theta_{t + 1} = \theta_t - \alpha \frac{m_t}{\sqrt{||v_t||}}$.
> > > The effective update magnitude is $\alpha\frac{||m_t||}{\sqrt{||v_t||}}$. This coupling of first-order momentum with a noise-sensitive second-moment estimate makes it hard to control $\alpha$ without smoothness or variance parameters, hindering any problem-parameter-free convergence guarantees, especially in MAML settings.
> > > (Existing results for related methods (e.g., Theorem 5 on AMSGrad in Yang et al., 2023) only cover highly simplified cases (e.g., $\beta_1=\beta_2=0$).)
> > >
> > > - __Why TFMAML admits a problem-parameter-free analysis.__
> > > Our update
> > > $m_t = \gamma m_{t-1} + (1-\gamma) g_t$ and update as $\theta_{t + 1} = \theta_t - \beta \frac{m_t}{||m_t||}$
> > > uses _only_ first-order momentum and explicit normalization. The normalization fixes the update magnitude to exactly $\beta$, making the stepsize independent of gradient scale, smoothness constants, etc. This yields a stable update rule whose behavior can be analyzed cleanly, established in Theorem 1 and Appendix B.
> > >
> > > __Reference:__
> > > __(Yang et al., 2023)__ Yang, J., Li, X., Fatkhullin, I., & He, N. (2023). Two sides of one coin: the limits of untuned SGD and the power of adaptive methods. Advances in Neural Information Processing Systems, 36, 74257-74288.

---

### Author Response · Authors · 2025-12-02
**Response to Reviewers and Summary of Revisions**

Dear Reviewers and Area Chair,


We sincerely thank you for the valuable feedback and constructive suggestions. Below we summarise the key discussion points and the major improvements incorporated into the revision.

Our work proposes a problem-parameter-agnostic MAML algorithm achieved through a simple yet effective combination of gradient normalization and momentum. This design yields __stepsize tuning-free convergence guarantees without any non-diminishing terms__ and __without requiring smoothness or curvature parameters__, which are typically unavailable in practice. To our knowledge, this is the __first convergence result for MAML/FOMAML that is fully free of problem-specific parameters__. We made the following improvement during the discussion phase:

- __Empirical Enhancements (nLZX, ejKG)__

    1. __Broader validation:__ We added experiments on __MiniImageNet__ with a __different model architecture__ from that used on Omniglot, further demonstrating the __robustness and superiority__ of our method across datasets and networks.
    2. __Momentum effect visualization:__ We included a comparison of __gradient-norm trajectories__, clearly illustrating how momentum (vs. pure SGD or pure normalization) __reduces gradient bias__ and __accelerates convergence__.

- __Theoretical Clarifications and Strengthening__
    1. __Role of momentum in bias reduction (nLZX):__ We now explicitly show how momentum applies an __summed exponential moving average__ to the error terms, thereby diminishing bias.
    2. __Clear differentiation from Adam, Cutkosky’s notes, and Fallah’s framework (nLZX, ejKG, PtYt)__: We explain why existing adaptive methods do not yield parameter-free convergence for MAML, and how our approach overcomes these limitations by delivering the __first theoretically-guaranteed problem-parameter-agnostic MAML/FOMAML__ with __no non-vanishing residual terms__.
    3. __Clarified MAML-specific formulation (nLZX):__ We emphasize the __MAML (not multitask learning)__ framework throughout the theory, algorithm, and experiments, highlighting the structural differences and implications.

All revisions are marked in blue in the updated manuscript.
We sincerely thank the reviewers for the insightful comments and the Area Chair for overseeing the process. Thank you again for your time and consideration.

Best regards,

Authors of the submission 10838

---

### Meta-Review · Area_Chair_CXw8 · 2026-01-05

**Summary:**

The reviewers agree that the method is well-motivated and solves an important practical problem in MAML, namely the need for problem-specific hyper-parameter tuning. The proposed method is well motivated theoretically and easy to implement practically. The key weaknesses identified include unclear positioning with respect to prior work, the limited experimental setting of using only Omniglot, and the lack of clear explanation of how momentum removes bias.

**Reviewer Concerns:**

Among the key concerns, the authors addressed the limited experimental setting by including new experiments on MiniImageNet. The authors also clarified in their rebuttal how momentum alleviates the issue of bias in gradient estimation in MAML. Lastly, the authors have improved upon the positioning of the paper with respect to earlier work.

**Reviewer Scores:**

While I do not foresee major changes in the score of the positive reviewers, the negative reviewer's primary concern of improving clarity and additional experiments have been partially addressed, and is hence expected to slightly improve their evaluation. Overall, the paper remains borderline. I regard the primary contribution of the paper to be an algorithmic one that is well-supported by theory, but I think the paper may benefit from further iterations on two fronts: 1) greatly expand the experimental setup to demonstrate across more than 2 datasets and architectures, since the primary innovation of parameter-free tuning, this has not been thoroughly demonstrated; 2) tighter coupling between the theory and the experiments, e.g. having experiments quantitatively demonstrate the rate estimates in Theorems 1 and 2. The current experiments are indirect, qualitative observations that are consistent with convergence behaviours, but do not confirm the theory in a direct way.

---

### Decision · Program_Chairs · 2026-01-26

Reject